# OFFLINE IMITATION LEARNING WITHOUT AUXILIARY HIGH-QUALITY BEHAVIOR DATA

## ABSTRACT

In this work, we study the problem of Offline Imitation Learning (OIL), where an agent aims to learn from the demonstrations composed of expert behaviors and sub-optimal behaviors without additional online environment interactions. Previous studies typically assume that there is high-quality behavioral data mixed in the auxiliary offline data and seriously degrades when only low-quality data from an off-policy distribution is available. In this work, we break through the bottleneck of OIL relying on auxiliary high-quality behavior data and make the first attempt to demonstrate that low-quality data is also helpful for OIL. Specifically, we utilize the transition information from offline data to maximize the policy transition probability towards expert-observed states. This guidance can improve long-term returns on states that are not observed by experts when reward signals are not available, ultimately enabling imitation learning to benefit from low-quality data. We instantiate our proposition in a simple but effective algorithm, Behavioral Cloning with Dynamic Programming (BCDP), which involves executing behavioral cloning on the expert data and dynamic programming on the unlabeled offline data respectively. In the experiments on benchmark tasks, unlike most existing offline imitation learning methods that do not utilize low-quality data sufficiently, our BCDP algorithm can still achieve an average performance gain of more than 40% even when the offline data is purely random exploration.

## 1 INTRODUCTION

The recent success of offline Reinforcement Learning (RL) in various fields demonstrate the significant potential of addressing sequential decision-making problem in a data-driven manner (Levine et al., 2020). Offline RL enables the learning of policies from logged experience, reducing the reliance on online interactions and making RL more practical, especially when online data collection may be expensive or risk-sensitive (Sinha et al., 2021; Qin et al., 2022; Fang et al., 2022). However, in many real-world applications, offline RL encounters two major challenges: quantity-quality dilemma on logged data and tricky design of reward function. In offline RL tasks, high-quality expert data is often expensive and difficult to obtain, and it is usually difficult to demand both quantity and quality from offline data (Levine et al., 2020; Liu et al., 2021). Moreover, the reward function, which determines the desired behavior of the agent, usually needs to be custom-designed for each task. This requires sufficient prior knowledge and can be challenging in fields such as robotics (Bobu et al., 2022), autonomous driving (Knox et al., 2023), and healthcare (Yu et al., 2023). One popular paradigm for breaking these practical barriers is *Offline Imitation Learning*, which trains an agent using limited expert demonstrations and reward-free logged data from arbitrary polices.

Recent offline imitation learning (IL) methods have achieved promising success that benefits from unlabeled offline data of uncertain quality. For example, DemoDICE (Kim et al., 2022) extends adversarial imitation learning by executing state-action distribution matching on offline data as a regularization term. DWBC (Xu et al., 2022a) regards the offline data as a mixture of expert data and suboptimal data and employs positive-unlabeled learning to build a discriminator that can identify these expert-similar behaviors from offline data. OTIL (Luo et al., 2023) utilizes optimal transport to discover an alignment that has the least Wasserstein distance between unlabelled trajectories and expert demonstrations. This similarity measure is then used to provide reward annotation and off-the-shelf offline RL algorithms are applied to learn the agent.

All of the above positive results, however, are based on a basic assumption that there is some high-quality behavioral data mixed in the auxiliary offline data that is similar to expert trajectories. Such an assumption may differ from the real situation because the offline data may come from any off-policy distribution, which can be targeted towards different goals, suboptimal policies, or even random exploration (Levine et al., 2020). Unfortunately, existing offline imitation learning struggles with these low-quality auxiliary data (Li et al., 2023b). In other words, offline imitation learning cannot benefit from them and may even underperform a baseline that does not use auxiliary data, as illustrated in Figure 1. Such phenomena undoubtedly go against the expectation of offline imitation learning and limit its effectiveness in a large number of practical tasks. This naturally leads to the following question:

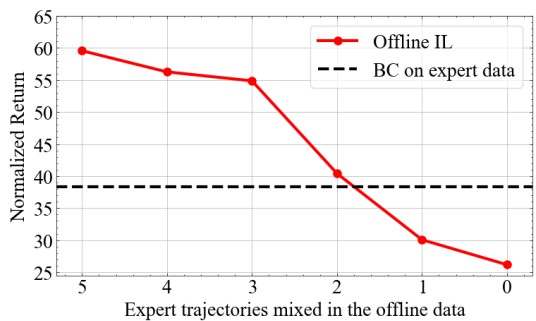

Figure 1: The performance of offline imitation learning degrades significantly as the number of expert trajectories in the unlabeled data decreases. The experimental details are provided in the C.2.

*Could Offline Imitation Learning benefit from auxiliary low-quality behavior data?*

In this work, we demonstrate that this is actually achievable, breaking through the previous assumptions about the behavior quality of offline policy. Unlike the existing offline imitation learning, we do not search for behaviors in the offline data that are similar to the expert data, which is futile when dealing with low-quality behaviors. Instead, we utilize the transition information in the offline data to guide the policy toward expert-observed states. This guidance can improve long-term returns on states that are not observed by experts when reward signals are also not available, ultimately enabling imitation learning to benefit from low-quality behavior data. We instantiate our proposition in a simple but effective algorithm, Behavioral Cloning with Dynamic Programming (BCDP), which involves executing behavioral cloning on the expert data and dynamic programming on the unlabeled offline data separately. On the one hand, imitation of expert data can obtain behavioral rewards on the expert-observed states; on the other hand, dynamic programming on offline data increases the likelihood of transitioning to expert-observed states and improves the agent's ability in expert-unobserved states. In the experiments on D4RL benchmark tasks, unlike most existing offline imitation learning methods that do not utilize low-quality data sufficiently, our BCDP algorithm can still achieve an average performance gain of more than 40% even when the offline data is purely random exploration.

## 2 BRIEF INTRODUCTION TO OFFLINE IMITATION LEARNING

This section provides a brief review of offline imitation learning, including the problem formulation and the two main branches of solutions: behavioral cloning and inverse reinforcement learning.

### 2.1 BACKGROUND AND FORMULATION

In this work, we consider the infinite Markov Decision Process (MDP) setting (Sutton et al., 1998), denoted as $\mathcal{M} = \{\mathcal{S}, \mathcal{A}, T, r, d_0, \gamma\}$. Here, $\mathcal{S}$ is the state space, $\mathcal{A}$ is the action space, $T : \mathcal{S} \times \mathcal{A} \to S$ is the transition probability of $\mathcal{M}$, $r : \mathcal{S} \times \mathcal{A} \to [0, 1]$ is the reward function, $d_0 : \mathcal{S} \to \Delta(S)$ is the initial state distribution and $\gamma \in (0, 1)$ is the discount factor. The decision-making process occurs sequentially. At time $t$, the agent observes a state $s_t \in S$ and takes an action $a_t$, following the conditional probability $\pi(a_t|s_t)$. The agent then receives a reward $r(s_t, a_t)$ from the environment, and a new state $s_{t+1}$ appears based on the transition probability $T(s_{t+1}|s_t, a_t)$ of $\mathcal{M}$. The goal of sequential decision-making is to maximize the expected cumulative reward:

$$J(\pi) = \mathbb{E}_{s_0 \sim d_0, s_{t+1} \sim T(\cdot|s_t, \pi(s_t))} \left[ \sum_{t=0}^{\infty} \gamma^t r(s_t, \pi(s_t)) \right]. \tag{1}$$

In the offline imitation learning setting, there are limited expert demonstrations with $N_E$ trajectories:

$$D^E = \{(s_0, a_0, s_1, a_1, \ldots, s_h, a_h) | s_0 \sim d_0, a_t \sim \pi^E(\cdot | s_t), s_{t+1} \sim T(\cdot | s_t, a_t), \forall t \in [h]\}$$

where $h$ is the length of each trajectory following the expert policy. There are also some logged experiences collected by the arbitrary behavior policies $\pi$ which is much cheaper to obtain:

$$D^O = \{(s_0, a_0, s_1, a_1, \ldots, s_h, a_h) | s_0 \sim d_0, a_t \sim \pi(\cdot | s_t), s_{t+1} \sim T(\cdot | s_t, a_t), \forall t \in [h]\}.$$

In the literature (Levine et al., 2020), the offline policy $\pi$ can come from various task objectives, suboptimal policies, or even random exploration, which are different from the expert policy and thus challenging to utilize in the imitation learning.

## 2.2 BEHAVIORAL CLONING

Behavior Cloning is a classical imitation learning method which optimizes the policy via supervised learning (Pomerleau, 1988). Recent studies provide a generalized objective:

$$\max_{\pi} \frac{1}{|D^E|} \sum_{(s,a) \in D^E} \log \pi(a|s) + \alpha \frac{1}{|D^O|} \sum_{(s,a) \in D^O} \log \pi(a|s) \cdot f(s,a). \tag{2}$$

where $f : \mathcal{S} \times \mathcal{A} \rightarrow [0, 1]$ and $\alpha$ balances the utilization of offline data. The behavioral cloning maximizes the log-likelihood on the empirical observations. Recently, DemoDICE (Kim et al., 2022) takes the $\alpha$ regularization to provide proper policy regularization. DWBC (Xu et al., 2022a), set the $f(s, a) = \frac{d_E^\pi(s,a)}{d_O^\pi(s,a)}$ and then implement an importance sampling via an distribution discriminator. As a result, the expert-similar behavior will be filtered. ORIL (Zolna et al., 2020) trains a critic network $Q$ to evaluate actions $(s, a)$ and sets $f(s, a) = \mathbb{I}[Q(s, a) > Q(s, \pi(s))]$ to eliminate actions that are considered inferior to the current policy $\pi$. Sasaki & Yamashina (2021) use the old policy $\pi'$ during the learning process to weight $f(s, a) = \pi'(a|s)$, thus eliminates the noisy demonstrations in the offline data. A delicate $f$ can help identify expert trajectories in the offline data and enhance the imitation towards the expert policy $\pi^E$. However, when the offline data does not contain expert trajectories but only sub-optimal data or purely random explorations, behavioral cloning cannot benefit from these data and may even perform worse compared to the counterpart that only uses expert data.

## 2.3 INVERSE REINFORCEMENT LEARNING

Inverse Reinforcement Learning is another popular branch to implement reinforcement learning without reward supervision (Ng & Russell, 2000). It involves iteratively learning a reward function and policy (Ziebart et al., 2008; Arora & Doshi, 2021). However, this requires a potentially large number of online interactions during training, which can result in poor sample efficiency. Recently, offline inverse reinforcement learning has been proposed to eliminate the online interactions and learn from the offline demonstrations (Jarboui & Perchet, 2021; Luo et al., 2023). Cameron (Jarboui & Perchet, 2021) built a discriminator between expert demonstrations and offline demonstrations to serve as the reward function $\hat{r}$, which takes a similar underlying idea like discriminator-based imitation learning, that is, the state-action pairs $(s, a)$ which resemble expert data receive higher rewards. Luo et al. (2023) uses optimal transport to find an alignment with the minimal wasserstein distance between unlabeled trajectories and expert demonstrations. The similarity measure between a state-action pair in unlabeled trajectory and that of an expert trajectory is then treated as a reward label. Given the reward function, they further take a specific RL algorithm to learn a policy:

$$\max_{\pi} J_{RL}(\pi | D^E \cup D^O, \hat{r}), \hat{r}(s, a) = \text{similarity}((s, a), D^E). \tag{3}$$

When the offline data does not contain expert trajectories, the similarity evaluation becomes challenging in recovering the true reward. In some cases, it may even result in a zero reward, reducing it to a simple method known as unlabeled data sharing (UDS). UDS (Yu et al., 2022) simply applies zero rewards to unlabeled data and finds that it leads to effective utilization in offline reinforcement learning, even when the unlabeled data is incorrectly labeled. In this paper, we assign a zero reward for unlabeled data, similar to UDS. Unlike UDS which provides an explanation based on reward bias, we offer a new theoretical view for the effectiveness of offline imitation learning. In terms of practical method design, we made a modification on the top of offline RL method to adapt to the offline imitation learning, which led to a significant performance improvement in the experiments.

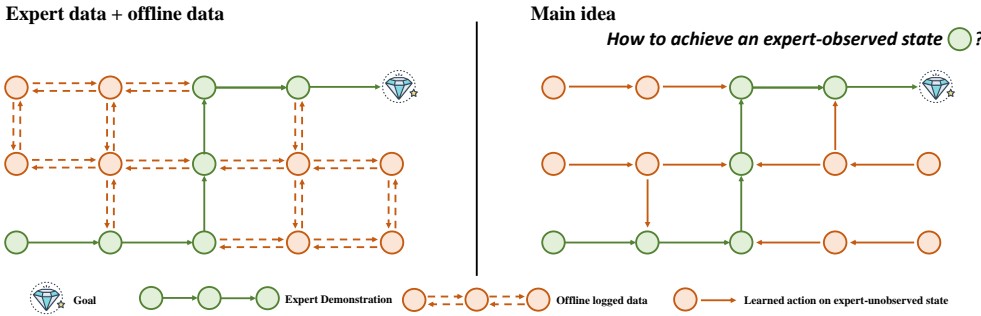

Figure 2: An illustration of our main idea: we make use of transition information in the offline data to guide the agent from expert-unobserved states to expert-observed states, ultimately ensuring a long-term return (reaching the target diamond).

## 3 THE PROPOSED METHOD

In the offline imitation learning setting, the reward labels for the offline data are unknown, leading to uncertain behavior quality. The agent can only rely on expert data to imitate the expert policy and maximize cumulative rewards. Fortunately, the transition information in the offline data is also reliable with the nature of sequential decision-making. We can leverage this information to guide policy optimization on expert-unobserved states. Figure 2 illustrates our main idea using the navigation as an example. Given limited expert demonstrations, the agent is confused when staying on a state (◯) has not been observed in the expert data. Nevertheless, we can guide the agent to the expert-observed states (◯) and then ensure that it reaches the goal (◈) via imitating expert behavior on the observed states. Based on this idea, we further present a solution which maximize the transition probability towards expert-observed states and the corresponding analysis.

### 3.1 LONG-TERM RETURN ON EXPERT-UNOBSERVED STATES

Following our main idea, we first formally present it with the standard MDP. The goal of sequential decision-making is to maximize a long-term cumulative return. Given a policy $\pi$, the value function on state $s$ at time $\mu$ could be formulated as:

$$V^\pi(s) = \sum_{a \in A} \pi(a|s) \cdot \left( \underbrace{r(s,a)}_{\text{one-step reward}} + \gamma \underbrace{\sum_{s' \in S} V^\pi(s')T(s'|s,a)}_{\text{long-term return}} \right) \quad (4)$$

Although the one-step reward $r(s,a)$ is uncertain in the context of offline imitation learning, the long-term return on the following states $s'$ can be guaranteed if $s'$ belongs to expert data $D^E$, when we follow the behavioral cloning on the expert-observed states, there is a lower bound (Rajaraman et al., 2020; Xu et al., 2022b):

**Lemma 1** (Lower Bound on Expert-Observed States). *For deterministic expert policy and behavioral cloning policy $\pi^E, \pi \in \Pi_{det}$ For any expert-observed state $s \in D^E$, the policy $\pi$ following behavioral cloning has a lower bound:*

$$V^\pi(s) \geq V^{\pi^E}(s) - \frac{1}{d_0(s)(1-\gamma)^2} \frac{4|S|}{9N^E} \quad (5)$$

At an expert-unobserved state $s \notin D^E$, if we find an action $a$ that could lead the agent to an expert-observed state, i.e., $\exists (s,a,s') \in D^O, s' \in D^E$, then we could ensure the long-term return of the action $a$ via the above lower bound. A formal proposition to maximize the distribution of expert-observed states could be provided as:

**Proposition 1** (Expert-State Distribution Maximization). *Given expert dataset $D^E$ and offline dataset $D^O$, the policy could imitate the expert behavior on the expert-observed states $s \in D^E$ and take the transition information in offline data to maximize the policy-dependent transition probability towards expert-observed states on the expert-unobserved states.*

$$\pi(s) = \begin{cases} \pi^E(s), & s \in D^E \\ \arg\max_{\pi'} \sum_{t=1}^{\infty} \gamma^t Pr(s_t = s' | \pi, s_0 = s) \cdot \mathbb{I}[s' \in D^E], & s \notin D^E \end{cases} \quad (6)$$

*Supposing the reward of expert behavior $\pi^E(s)$ is as least $R_E$, i.e., $r(s, \pi^E(s)) \geq R_E, \forall s \sim d^{\pi^E}$, this proposition could maximize a lower bound of policy $\pi$ as:*

$$J(\pi) \geq \sum_{t=0}^{\infty} \gamma^t \sum_{s \in D^E} Pr(s_t = s | \pi) \cdot R_E = \frac{1}{1 - \gamma} \sum_{s \in D^E} d^{\pi}(s) \cdot R_E \quad (7)$$

The above proposition proposes to maximize the exception of keeping the agent at expert-observed states in order to achieve an improvement in the lower bound of long-term returns. In this proposition, we assume that the behavior quality of expert policy is guaranteed, which is common in sequential decision-making. For examples, navigation experts are often able to clearly identify goals and provide the most efficient route to reach them. Similarly, in autonomous driving, experts are skilled human drivers who can ensure the safe operation of the vehicle. The equation 7 offers a straightforward and easy-to-understand lower bound. Based on the above lower bound, maximizing the expert-state distribution $\sum_{s \in D^E} d^{\pi}(s)$, i.e., lead the agent to the expert-visited states, could improve the lower bound of cumulative return $J(\pi)$. The form of our analysis is different from the previous imitation learning work (Rajaraman et al., 2020; Xu et al., 2022b;c). Most of them focus on minimizing the imitation gap $|J(\pi) - J(\pi^E)|$, which means imitating expert strategies as closely as possible, including making $d^{\pi}$ as close to $d^{\pi^E}$ as possible. In contrast, we choose to maximize $\sum_{s \in D^E} d^{\pi}(s)$ when the quality of expert behavior is guaranteed. The difference here is that our actions may not be consistent with those of the expert for unobserved states. However, please note that there is no direction for imitating experts on these expert-unobserved states, if only empirical observations are given. In contrast, we provide a guarantee of long-term returns via maximizing the expert-state distribution , and thus improve the lower bound.

## 3.2 PRACTICAL IMPLEMENTATION

To implement the proposition, we combine the dynamic programming with behavioral cloning to boost the long-term return on expert-unobserved states while imitating the expert behavior on the expert-observed states. Specifically, we choose the TD3 (Fujimoto et al., 2018) as the dynamic programming algorithm. TD3 builds network to estimate the Q-function of state-action pair:

$$\arg\min_Q \sum_{(s,a,s') \sim D^O \cup D^E} \|\mathcal{B}_{\gamma}^{\pi} Q(s,a) - Q(s,a)\|^2,$$

$$\mathcal{B}_{\gamma}^{\pi} Q(s,a) = r(s,a) + \gamma \sum_{a' \in A} \pi(a'|s') Q(s',a') \quad (8)$$

where $\mathcal{B}^{\pi}$ is the Bellman operator and $r(s,a) = \mathbb{I}[(s,a) \in D^E]$ with the indicator function $\mathbb{I}[\cdot]$. Then the policy $\pi$ is optimized via a objective:

$$\max_{\pi} \sum_{(s,a) \sim D^E} [\log \pi(a|s)] + \alpha \sum_{s \sim D^O \cup D^E} Q(s, \pi(s)) \quad (9)$$

The Equation 9 contains behavioral cloning and Q-learning to form our proposal, which involves imitating expert behavior on expert-observed states and ensuring long-term rewards through dynamic programming on expert-unobserved states. The overall implementation is summarized in the algorithm 1.

Our implementation makes a minimal derivation on the top of TD3+BC, a simple but effective offline reinforcement learning method (Fujimoto & Gu, 2021). The difference is that they do behavioral

---

**Algorithm 1** Behavioral Cloning with Dynamic Programming (BCDP)

---

**Require:** Expert data $D^E$, offline data $D^O$, Initialize critic network $Q_{\theta_1}, Q_{\theta_2}$ and actor network $\pi_\phi$ with random parameters $\theta_1, \theta_2, \phi$. Initialize delayed networks $\theta_1' \leftarrow \theta_1, \theta_2' \leftarrow \theta_2, \phi' \leftarrow \phi$.
  **for** $t = 1$ to $T$ **do**
    Sample mini-batches $b^E$ and $b^O$ of transitions $(s, a, s')$ from datasets $D^E, D^O$ respectively.
    $b^U \leftarrow b^E \cup b^O$
    $y \leftarrow \min_{i=1,2} \mathcal{B}_\gamma^{\pi_{\phi'}} Q_{\theta_i'}(s, a)$     ▷ Obtain the minimum estimation $y$ from delayed networks.
    $\theta_i \leftarrow 1/|b^U| \sum_{(s,a) \in b^U} \nabla_{\theta_i} (y - Q_{\theta_i}(s,a))^2$          ▷ Update Q-network via $y$.
    $G_{BC} \leftarrow 1/|b^E| \sum_{(s,a) \in b^E} \nabla_\phi - \log \pi(a|s)$    ▷ Calculate gradient with behavioral cloning.
    **if** $t \bmod t_{freq}$ **then**
      $G_Q \leftarrow 1/|b^U| \sum_{s \in b^U} \nabla_\phi - Q(s, \pi(s))$      ▷ Calcualte policy gradient with Q function
      $\theta_1' \leftarrow \tau\theta_1 + (1 - \tau)\theta_i'$                    ▷ Update the delayed networks.
      $\theta_2' \leftarrow \tau\theta_2 + (1 - \tau)\theta_i'$
      $\phi' \leftarrow \tau\phi + (1 - \tau)\phi'$
    **end if**
    $\phi \leftarrow \phi - \eta(G_{BC} + \alpha G_Q)$           ▷ Update actor network with learning rate $\eta$.
  **end for**

---

cloning on the entire offline data, while we do behavioral cloning on expert data. This difference actually indicates two completely different purposes. That is, TD3+BC's behavioral cloning is for conservatism, to keep the learned policy close to the offline policy and eliminate the estimation bias of Q-learning, while we hope to supplement dynamic programming on unseen states with behavioral cloning on expert data. In the offline imitation learning, applying behavior regularization on the unlabeled data without quality assurance may be unsafe and lead to fitting low-quality behaviors, thus harming the performance. In this paper, we have not added any additional regularization terms to maintain conservatism as traditional offline RL methods (Kumar et al., 2020; Fujimoto & Gu, 2021; Kostrikov et al., 2022), but have instead kept our approach simplified. We mainly focus on boosting the performance on non-visited states of expert data to improve offline imitation learning, and leave the development of these offline technologies to future work.

### 3.3 CONNECTION WITH THE RELATED WORK

Our study is motivated by the recent empirical success in offline IL. We find that most of them consider the integration of expert data into sub-optimal data or introduce noise into expert data to construct an offline dataset (Sasaki & Yamashina, 2021; Kim et al., 2022; Xu et al., 2022a; Li et al., 2023a), and then identify expert-similar data to improve IL. We argue that this assumption may not hold in real applications, and it also differs significantly from the existing offline RL literature (Levine et al., 2020). In this paper, we focus on the utilization of pure sub-optimal data replay (even random exploration), which is more in line with the objective of offline RL: improving performance using suboptimal data. For offline IL, we overcome the obstacle of not having learning guidance in expert-unobserved states, ensuring the cumulative return of the policy. In the field of online IL, SQIL (Reddy et al., 2020) also considers guiding the agent to expert-observed states to improve online sampling efficiency. BC-SAC (Lu et al., 2022) combines imitation and RL to improve the safety and reliability of autonomous driving. Unlike them, we demonstrate that BCDP can benefit from low-quality offline data. Utilizing low-quality data in offline imitation learning as a supplement to limited expert data is crucial, while in online imitation learning, agents can actively explore the environment and rarely collect a lot of low-quality data. Additionally, BCDP improves the lower bound of cumulative return and provides a promising way for offline imitation learning.

### 4 EMPIRICAL STUDY

The main objectives of our empirical study are to answer three questions: (1) How does BCDP perform relative to other offline IL methods? (2) How does the performance of BCDP agents vary as a function of the size of the expert and offline datasets, especially when the expert data is extremly scarce? (3) How do the BCDP agents exactly perform on the expert-unobserved states?

We conduct the experiments on the D4RL benchmark (Fu et al., 2020), includes a series tasks of navigation, locomotion and manupulation. We respond to the above questions in Section 4.1, 4.2 and 4.3, respectively. For all settings, we obtain undiscounted average returns of the policy at the last 10 evaluations of training. The average and deviation under three different seeds are reported.

## 4.1 Main Evaluation (Q1)

We evaluate BCDP on a wide range of domains in the D4RL benchmark (Fu et al., 2020).

**Navigation.** We conduct the experiments on the *Maze2D* environments to evaluate the policy. The *Maze2D* domain requires an agent to navigate in the maze to reach a fixed target goal and stay there. The D4RL benchmark provides three maze layouts (i.e., umaze, medium, and large) and two rewards types (i.e., sparse and dense reward singals) in this domain. We employs 5 expert trajectories as the expert data which follows a path planner Fu et al. (2020). We consider two types of offline data: randomly walking (umaze-random, medium-random, etc.) and logged experience with random goals (umaze-v1, medium-v1, etc.). The former considers the ability of offline imitation learning methods to use low-quality policy data, while the latter considers the ability to use related task data.

**Locomotion.** We conduct the experiments on the *Gym-MuJoCo* environments to evaluate the policy. It consists four different environments (i.e., hopper, walker2d, halfcheetah and ant). We employs 5 expert trajectories from the "-expert" dataset as the expert data. We also consider two types of offline data: random policy (hopper-random-v2, halfcheetah-random-v2, etc.) and sub-optimal policy (hopper-medium-v2, halfcheetah-medium-v2, etc.). The former contains logged experiences from a random policy, while the latter comes from an early-stopped SAC policy.

**Manupulation.** We conduct the experiments on the *Adroit* environments to evaluate the policy. *Adroit* (Rajeswaran et al., 2018) involves controlling a 24-DoF simulated Shadow Hand robot tasked with hammering a nail, opening a door, twirling a pen, or picking up and moving a ball. It measures the effect of a narrow expert data distributions and human demonstrations on a high-dimensional robotic manipulation task. We employs 50 expert trajectories from the "-expert-v1" dataset as the expert data. Two types of offline data are considered: human demonstrations (pen-human-v1, door-human-v1, etc.) and sub-optimal policy (pen-cloned-v1, door-cloned-v1, etc.). The former contains logged experiences from real humen, while the latter comes from an imitation policy.

**Competing Baselines.** We compare BCDP with the well-validated baselines and state-of-the-art offline imitation learning methods, includes: **BC-exp** and **BC-all** perform behavior cloning on expert data and union data, respectively. **DemoDICE** (Kim et al., 2022) and **DWBC** (Xu et al., 2022a) are two state-of-the-art methods based on the generalized behavioral cloning. **OTIL** (Luo et al., 2023) is a recently proposed offline inverse reinforcement learning method. UDS (Yu et al., 2022) labels all rewards from the unlabeled datasets with 0 and utilizes offline reinforcement learning algorithms to train the agent on the merged dataset. We have selected TD3+BC as our most similar offline RL algorithm, which allows it to be considered as an ablation study of our approach.

**Experimental Results.** We evaluate BCDP in fourteen D4RL benchmark domains with 28 settings. As shown in the Table 1, BCDP significantly outperforms baselines and achieves the best performance on 17 of 28 continuous control tasks. From the results, we have the following observations and analyses: The BC-exp baseline indicates the diffculty of imitating learning when the expert data is limited. The BC-all method uses all of the offline data for behavior cloning, without considering the possibility that some of the unlabelled data in the offline IL setting may come from low-quality policies that are not suitable for direct use in behavior cloning, leading to weaker performance compared to BC-exp in most cases. DemoDICE achieved good performance in offline data with high-quality behavior, such as locomotion tasks with medium settings. However, its performance is weak when the behavior quality is low, revealing that using the entire offline data as regularization for imitation learning makes it difficult to leverage the low-quality behavior data. DWBC uses positive-unlabeled learning to identify samples similar to expert behavior, which can help improve the performance of behavior cloning in situations where the data quality is low. It performs well in different settings. OTIL and UDS adopt the perspective of inverse reinforcement learning, which involves labeling offline data before conducting offline RL. We can observe that OTIL and UDS perform unsatisfactorily when the quality of offline data is low. One plausible reason is that, in offline RL, algorithms typically constrain the policy to be close to the offline distribution to avoid the risk of OOD exploration. However, in offline imitation learning, when the quality of the behavior data is

Table 1: The results on D4RL benchmark. All values are normalized to lie between 0 and 100, where 0 corresponds to a random policy and 100 corresponds to an expert policy (Fu et al., 2020). The best result in each setting is in bold and the second-best result is underlined.

| | Dataset | BC-exp | BC-all | DemoDICE | DWBC | OTIL | UDS | BCDP |
|---|---|---|---|---|---|---|---|---|
| **Navigation** | sparse-umaze-random | 88.9 ± 42.0 | 9.53 ± 14.5 | 81.0 ± 17.1 | 125. ± 19.1 | 33.6 ± 22.8 | -2.1 ± 11.5 | **126. ± 19.9** |
| | sparse-medium-random | 38.3 ± 18.1 | -2.8 ± 3.61 | 27.5 ± 8.58 | 26.1 ± 1.48 | 22.2 ± 14.5 | 19.5 ± 8.41 | **82.3 ± 28.3** |
| | sparse-large-random | 1.45 ± 6.63 | 5.92 ± 14.6 | 7.89 ± 7.76 | 13.1 ± 14.0 | 1.88 ± 3.19 | 47.8 ± 46.7 | **138. ± 19.0** |
| | dense-umaze-random | 61.7 ± 30.8 | 0.77 ± 14.0 | 66.9 ± 12.5 | 85.9 ± 5.34 | 7.27 ± 18.9 | -10. ± 8.01 | **89.2 ± 20.5** |
| | dense-medium-random | 37.6 ± 19.2 | -7.7 ± 4.08 | 27.0 ± 9.98 | 25.8 ± 5.78 | 16.9 ± 12.0 | 16.3 ± 15.9 | **72.4 ± 27.2** |
| | dense-large-random | 41.3 ± 53.2 | 6.92 ± 14.0 | 18.6 ± 9.00 | 16.8 ± 9.78 | 11.7 ± 11.2 | 39.9 ± 34.0 | **122. ± 15.3** |
| | sparse-umaze-v1 | 88.9 ± 42.0 | 47.1 ± 13.0 | 15.7 ± 1.66 | 128. ± 14.5 | 35.8 ± 7.35 | 91.1 ± 22.9 | **132. ± 22.0** |
| | sparse-medium-v1 | 38.3 ± 18.1 | 5.55 ± 7.89 | 24.4 ± 7.63 | 80.4 ± 16.4 | 88.5 ± 30.5 | 97.0 ± 20.0 | **137. ± 11.4** |
| | sparse-large-v1 | 1.45 ± 6.63 | 23.7 ± 21.4 | 60.7 ± 30.6 | **161. ± 43.7** | 50.6 ± 41.5 | 134. ± 26.0 | 124. ± 22.0 |
| | dense-umaze-v1 | 61.7 ± 30.8 | 31.8 ± 10.5 | 19.3 ± 2.95 | 95.7 ± 12.6 | 31.7 ± 7.36 | 75.4 ± 23.3 | **96.0 ± 16.1** |
| | dense-medium-v1 | 37.6 ± 19.2 | 31.0 ± 3.81 | 34.7 ± 7.26 | **98.9 ± 33.4** | 65.6 ± 3.96 | 88.8 ± 20.9 | 93.2 ± 23.5 |
| | dense-large-v1 | 41.3 ± 53.2 | 37.9 ± 20.7 | 68.6 ± 4.95 | **164. ± 50.8** | 84.4 ± 26.2 | 128. ± 18.7 | 110. ± 28.1 |
| **Locomotion** | hopper-random-v2 | 52.1 ± 6.76 | 2.31 ± 0.41 | 15.8 ± 3.62 | 63.2 ± 8.77 | 21.6 ± 3.23 | 1.15 ± 0.46 | **73.2 ± 9.66** |
| | halfcheetah-random-v2 | 15.3 ± 7.36 | 2.25 ± 0.00 | 2.20 ± 0.01 | 13.3 ± 4.65 | 2.25 ± 0.00 | 4.62 ± 1.15 | **18.8 ± 6.11** |
| | walker2d-random-v2 | 105. ± 3.53 | 2.06 ± 2.34 | 30.4 ± 5.22 | 104. ± 6.03 | 7.43 ± 0.23 | -.11 ± 0.00 | **105. ± 2.08** |
| | ant-random-v2 | 41.0 ± 7.17 | 52.4 ± 8.41 | **55.5 ± 13.4** | 51.3 ± 6.88 | 31.2 ± 0.09 | 30.4 ± 2.99 | 54.1 ± 9.01 |
| | hopper-medium-v2 | 52.1 ± 6.76 | 56.4 ± 1.86 | 52.5 ± 1.19 | 74.5 ± 11.2 | 62.5 ± 25.9 | 66.0 ± 0.49 | **98.7 ± 5.81** |
| | halfcheetah-medium-v2 | 15.3 ± 7.36 | 42.8 ± 0.41 | 40.6 ± 1.50 | 16.2 ± 5.52 | 34.7 ± 1.82 | **57.1 ± 6.91** | 18.4 ± 10.9 |
| | walker2d-medium-v2 | **105. ± 3.53** | 86.8 ± 5.28 | 75.1 ± 1.61 | 77.8 ± 14.3 | 79.6 ± 1.70 | 8.52 ± 4.99 | 98.0 ± 1.94 |
| | ant-medium-v2 | 41.0 ± 7.17 | **98.7 ± 3.68** | 90.0 ± 3.18 | 41.9 ± 10.7 | 96.4 ± 2.32 | 18.4 ± 10.5 | 59.7 ± 16.2 |
| **Manipulation** | pen-human-v1 | 68.6 ± 35.0 | 51.9 ± 16.0 | **106. ± 22.3** | 50.5 ± 10.3 | 97.3 ± 25.7 | 8.56 ± 15.4 | 91.1 ± 20.8 |
| | door-human-v1 | 5.25 ± 7.90 | 9.30 ± 8.58 | 9.30 ± 8.58 | 1.04 ± 1.36 | **99.7 ± 2.78** | -.33 ± 0.01 | 4.16 ± 5.40 |
| | hammer-human-v1 | 101. ± 17.7 | 8.00 ± 5.89 | 28.5 ± 21.3 | 83.3 ± 14.0 | 66.0 ± 16.8 | 3.07 ± 0.06 | **109. ± 16.0** |
| | relocate-human-v1 | **59.8 ± 32.9** | 10.5 ± 5.23 | 1.31 ± 2.07 | 47.4 ± 12.8 | 41.7 ± 4.69 | -.34 ± 0.06 | 41.1 ± 15.4 |
| | pen-cloned-v1 | 68.6 ± 35.0 | 5.89 ± 8.01 | 33.1 ± 10.9 | 75.6 ± 27.2 | 58.7 ± 27.5 | 4.32 ± 8.03 | **103. ± 13.3** |
| | door-cloned-v1 | 5.25 ± 7.90 | 0.02 ± 0.04 | 0.07 ± 0.09 | 0.36 ± 0.33 | 0.29 ± 0.38 | -.33 ± 0.01 | **9.96 ± 8.94** |
| | hammer-cloned-v1 | 101. ± 17.7 | 0.28 ± 0.00 | 0.24 ± 0.01 | 98.6 ± 8.40 | 1.50 ± 0.86 | 0.38 ± 0.07 | **106. ± 21.3** |
| | relocate-cloned-v1 | **59.8 ± 32.9** | 10.5 ± 5.23 | -0.1 ± 0.09 | 56.2 ± 23.7 | -0.1 ± 0.04 | -.32 ± 0.03 | 34.7 ± 8.96 |

low, this may constrain the agent to low-quality policies and result in poor performance. Note that UDS, like our proposal, assigns a zero reward to offline data, which actually is an ablation case of our method. We found that our method, BCDP, has demonstrated significant performance improvement in the experiments. This is due to the fact that we treat the dynamic programming on offline data as an aid for behavior cloning on expert data, thereby avoiding excessive conservatism in offline RL. Moreover, BCDP ensures the performance of expert-observed states through behavior cloning on expert data while also providing guidance for dynamic programming on unlabeled data. Even though the offline data comes from pure random exploration, unlike existing methods that often fall short of the BC-exp baseline, BCDP achieves an average improvement of 43.6 (normalized score).

## 4.2 ABLATION STUDY FOR DATA-CENTRIC PROPERTIES (Q2)

To enable a systematic understanding of the BCDP, we vary the scale of expert data to examine the performance of BCDP. In the Figure 3, we present the comparison of BCDP with recent offline imitation learning methods, i.e., DemoDICE, OTIL and DWBC, for expert budgets ranging from 1 to 5 trajectories (from 10 to 50 trajectories in the Adroit settings). In the navigation task, there is always a transition path from any expert-unobserved states to expert-observed states, and our method is significantly better than existing offline imitation learning methods. In tasks such as locomotion and manipulation, the agent may fail to transfer to the expert-observed states, such as getting stuck in some difficult-to-recover states. Our method is also competitive in these tasks, showing the practical effectiveness of improving the lower bound of long-term return. In particular, when expert data is very scarce, BCDP can still make significant performance improvements by using dynamic programming to benefit from unlabeled data.

## 4.3 CONCERNS ON EXPERT-UNOBSERVED STATES (Q3)

To delve deeper into how BCDP improves imitating learning, we provide a detailed analysis on navigation tasks: *maze2d-medium-dense* and *maze2d-large-dense*, where obtaining long-term returns is crucial. Specifically, we sampled 1000 trajectories in medium environments and 500 trajectories in large environments based on the learned agents, and estimated the distance deduction and long-term return of their behaviors in different states, as shown in the Figure 4. We aim to quantitatively ana-

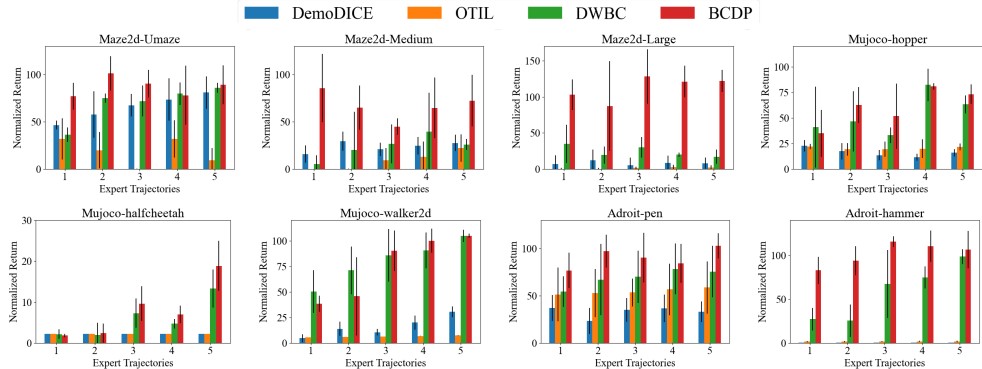

Figure 3: Comparision under varying number of expert trajectories.

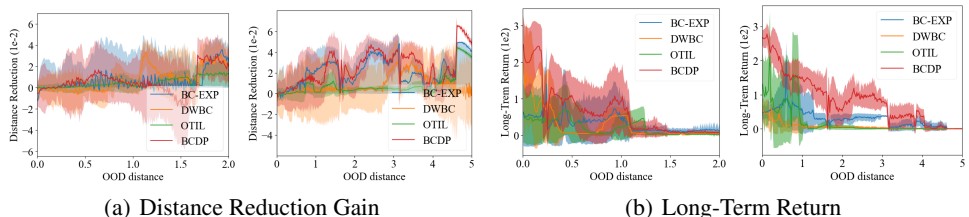

(a) Distance Reduction Gain

(b) Long-Term Return

Figure 4: Quantitative analysis of behavior in expert-unobserved states.

lyze whether BCDP has gained the ability to transfer to expert states in OOD states, and therefore define the following 1-NN-based measure to assess the improvement.

**Definition** (Distance Reduction Gain). *Given a expert dataset $D^E$ and a policy $\pi$, the excepted distance reduction on state $s$:*

$$\mathbb{E}[DRG(s)] = \min_{s_1 \in D^E} \|s - s_1\|_2 - \sum_{s'} T(s'|s,a)\pi(a|s) \cdot \min_{s_2 \in D^E} \|s' - s_2\|_2. \tag{10}$$

If the expected reduction on state DRG($s$) is greater than 0, it means that the next state $s'$ is closer to the expert-observed states relative to the current state $s$, corresponding to our proposal. In the Figure 4(a), we report the expected DRG for the different states $s$ with their distances from the expert data $D^E$. The OOD distance ($x$-axis) for state $s$ is calcualted as: $\min_{s_1 \in D^E} \|s - s_1\|^2$. We could find that the BCDP actually has a positive excepted DRG and tend to the expert-observed states. Their long-term return on the different states is also reported in the Figure 4(b). The results implicate that BCDP actually executes a conservate action to the expert-observed states and thus achieve more long-term return, especially on the states which is far from the expert data.

## 5 CONCLUSION

In this paper we tackle an important problem of offline imitation learning, that is, struggling with the low-quality auxiliary data from off-policy distribution. We show that the transition information in the offline data can be used to establish optimization objectives in expert-unobserved states and propose a simple but effective algorithm, BCDP (Behavioral Cloning with Dynamic Programming). Experiments demonstrate that BCDP can efficiently leverage low-quality behavior data and achieve state-of-the-art performance on the D4RL benchmark with 14 tasks. BCDP has made a step to break the behavior quality assumption of auxiliary data and extends the ability of offline imitation learning. One potential future direction is to extend BCDP with the model-based method, improving the efficiency of utilizing offline transition information. General theoretical formalization with the existing imitation gap in offline imitation learning is also an interesting future direction.

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

## A  PROOF OF LEMMA 1

**Lemma A.1** (Lower Bound on Expert-Observed States). *For deterministic expert policy and behavioral cloning policy $\pi^E, \pi \in \Pi_{det}$ For any expert-observed state $s \in D^E$, the policy $\pi$ following behavioral cloning has a lower bound:*

$$V^\pi(s) \geq V^{\pi^E}(s) - \frac{1}{d_0(s)(1-\gamma)^2}\frac{4|S|}{9N^E} \tag{11}$$

**Proof A.1.** *Following the Theorem 1 in the (Xu et al., 2022b),we first define the state-action distribution $d_s^\pi(\cdot,\cdot)$ of policy $\pi$ as: $d_s^\pi(\tilde{s},\tilde{a}) := \sum_{t=0}^\infty Pr(s_t = \tilde{s}, a_t = \tilde{a}|s_0 = 0, \pi)$.*

$$
\begin{aligned}
&V^{\pi^E}(s) - V^\pi(s) \\
=& \frac{1}{1-\gamma}\mathbb{E}_{\tilde{s},\tilde{a}\sim d_s^{\pi^E}(\cdot,\cdot)}r(\tilde{s},\tilde{a}) - \frac{1}{1-\gamma}\mathbb{E}_{\tilde{s},\tilde{a}\sim d_s^\pi(\cdot,\cdot)}r(\tilde{s},\tilde{a}) \\
\leq& \frac{1}{1-\gamma}\sum_{(s,a)\in\mathcal{S}\times\mathcal{A}}|d_s^{\pi^E}(s,a) - d_s^\pi(s,a)|\cdot 1 \\
=& \frac{1}{1-\gamma}\sum_{(s,a)\in\mathcal{S}\times\mathcal{A}}|(\pi^E(a|s) - \pi(a|s))d_s^{\pi^E}(s) + (d_s^{\pi^E}(s) - d_s^\pi(s))\pi(a|s)| \\
\leq& \frac{1}{1-\gamma}\left(\sum_{(s,a)\in\mathcal{S}\times\mathcal{A}}|(\pi^E(a|s) - \pi(a|s))d_s^{\pi^E}(s) + \sum_{s\in\mathcal{S}}|d_s^{\pi^E}(s) - d_s^\pi(s)|\right)
\end{aligned} \tag{12}
$$

*Following the Lemma 5 of (Xu et al., 2022b), the differernce on state distribution could be bounded by the poicy difference: $\sum_{s\in\mathcal{S}}|d_s^{\pi^E}(s) - d_s^\pi(s)| \leq \frac{\gamma}{1-\gamma}\mathbb{E}_{s\sim d_s^{\pi^E}}\sum_a|\pi^E(a|s) - \pi(a|s)|$. For the*

*deterministic policies $\pi$ and $\pi^E$, $\sum_a |\pi^E(a|s) - \pi(a|s)| = \mathbb{I}[s \neq D^E]$. Then we have a policy-difference-based upper bound:*

$$
\begin{aligned}
&V^{\pi^E}(s) - V^\pi(s) \\
&\leq \frac{1}{(1-\gamma)^2} \mathbb{E}_{s \sim d_s^{\pi^E}} \sum_a |\pi^E(a|s) - \pi(a|s)| \\
&= \frac{1}{(1-\gamma)^2} \sum_{\tilde{s}} d_s^{\pi^E}(\tilde{s}) \cdot Pr(\tilde{s} \notin D^E) \\
&= \frac{1}{(1-\gamma)^2} \sum_{\tilde{s}} d_s^{\pi^E}(\tilde{s}) \cdot (1 - d^{\pi^E}(\tilde{s}))^{|N_E|} \\
&= \frac{1}{(1-\gamma)^2} \sum_{\tilde{s}} \frac{d_s^{\pi^E}(\tilde{s})}{d^{\pi^E}(\tilde{s})} d^{\pi^E}(\tilde{s}) \cdot (1 - d^{\pi^E}(\tilde{s}))^{|N_E|} \\
&\leq \frac{1}{(1-\gamma)^2} \sum_{\tilde{s}} \frac{d_s^{\pi^E}(\tilde{s})}{\sum_{\hat{s}} d_0(\hat{s}) d_{\hat{s}}^{\pi^E}(\tilde{s})} d^{\pi^E}(\tilde{s}) \cdot (1 - d^{\pi^E}(\tilde{s}))^{|N_E|} \\
&\leq \frac{1}{(1-\gamma)^2} \sum_{\tilde{s}} \frac{d_s^{\pi^E}(\tilde{s})}{d_0(s) d_s^{\pi^E}(\tilde{s})} d^{\pi^E}(\tilde{s}) \cdot (1 - d^{\pi^E}(\tilde{s}))^{|N_E|} \\
&\leq \frac{1}{(1-\gamma)^2} \sum_{\tilde{s}} \frac{d_s^{\pi^E}(\tilde{s})}{d_0(s) d_s^{\pi^E}(\tilde{s})} d^{\pi^E}(\tilde{s}) \cdot (1 - d^{\pi^E}(\tilde{s}))^{|N_E|} \\
&= \frac{1}{(1-\gamma)^2} \frac{1}{d_0(s)} \sum_{\tilde{s}} d^{\pi^E}(\tilde{s}) \cdot (1 - d^{\pi^E}(\tilde{s}))^{|N_E|} \\
&\leq \frac{1}{d_0(s)(1-\gamma)^2} \frac{4|S|}{9 N_E}
\end{aligned}
\tag{13}
$$

*The least inequality comes from (Ross & Bagnell, 2010). The lower bound of the value function $V^\pi(s)$ could be concluded.*

## B  DISCUSSION WITH MODEL-BASED METHODS

In this paper, our main focus is on model-free methods in the imitation learning community. However, there are also some model-based methods. One such method is MobILE Kidambi et al. (2021), which addresses online imitation learning when expert actions are not available. It achieves this by integrating the idea of optimism in the face of uncertainty into the distribution matching framework. In offline imitation learning, MILE Hu et al. (2022) leverages 3D geometry from high-resolution videos of expert demonstrations and then plans entirely predicted in imagination to execute complex driving maneuvers. DMIL Zhang et al. (2022) introduces a discriminator to simultaneously distinguish the dynamics correctness and suboptimality of model rollout data against real expert demonstrations. MILO Chang et al. (2021) presents a model-based method to build a dynamic model ensemble on offline data and then solve a min-max imitation objective on the trajectories sampled by the dynamics model.

These model-based methods utilize the transition information to augment the trajectories for imitation learning. This is completely different from our goal. (1) On the one hand, since the data does not have reward labels, the trajectories sampled by model-based methods are still unlabeled. We still need to consider how to utilize these low-quality behavior data that have been sampled. Thus, the problem we presented in this paper is still unsolved. (2) On the other hand, these methods still perform distribution / similarity matching between the generated trajectories and the state-action distribution of expert behavior ($p(s, a)$) to annotate reward labels, which is consistent with our summary of existing work in equations 2 and 3. As we discussed in sections 2.2 and 2.3, they cannot provide optimization objectives on expert-unobserved states because this similarity can only rely on the model's function approximation ability to distinguish the quality of unlabeled trajectories and utilize them. This makes it difficult for them to effectively utilize offline data, which may contain many expert-unobserved states. On the contrary, when utilizing transition information, we consider

the subsequent expert-state distribution maximization. This approach does not expect the sampled trajectories to necessarily contain samples with a similar state-action distribution to the expert data. Instead, it provides guidance to the agent towards expert-observed states through the transition information between states, thereby improving the lower bound of long-horizon return.

## C  EXPERIMENTAL DETAILS

### C.1  DETAILS OF DATASETS

For navigation task, we generate offline data from random policy to evaluate the offline imitation learning with low-quality data. Specifically, we use the random agent provided by D4RL sample 5000 trajectories with 200 transitions in the corresponding environments. Random data in other settings are all provided by the D4RL, and we directly conduct the experiments on them. Table 3 provides a detailed statistics on all settings.

Table 2: Additional Details

| Setting | Expert Data | | Offline Data | |
|---|---|---|---|---|
| | #Transitions | Ave. of Rewards | #Transitions | Ave. of Rewards |
| sparse-umaze-random | 1490 | 0.561 | 1490000 | 0.094 |
| sparse-medium-random | 2990 | 0.711 | 2990000 | 0.028 |
| sparse-large-random | 3990 | 0.792 | 3990000 | 0.012 |
| dense-umaze-random | 1490 | 0.625 | 1490000 | 0.241 |
| dense-medium-random | 2990 | 0.738 | 2990000 | 0.073 |
| dense-large-random | 3990 | 0.819 | 3990000 | 0.046 |
| sparse-umaze-v1 | 1490 | 0.561 | 976725 | 0.080 |
| sparse-medium-v1 | 2990 | 0.711 | 1976410 | 0.023 |
| sparse-large-v1 | 3990 | 0.792 | 3966628 | 0.008 |
| dense-umaze-v1 | 1490 | 0.561 | 1490000 | 0.094 |
| dense-medium-v1 | 2990 | 0.738 | 1976410 | 0.065 |
| dense-large-v1 | 3990 | 0.819 | 3966628 | 0.035 |
| hopper-random-v2 | 4990 | 3.609 | 954757 | 0.832 |
| halfcheetah-random-v2 | 4990 | 10.743 | 998000 | -0.288 |
| walker2d-random-v2 | 4990 | 4.92 | 951090 | 0.091 |
| ant-random-v2 | 4456 | 4.393 | 993537 | -0.338 |
| hopper-medium-v2 | 4990 | 3.609 | 997719 | 3.108 |
| halfcheetah-medium-v2 | 4990 | 10.743 | 998000 | 4.77 |
| walker2d-medium-v2 | 4990 | 4.92 | 998128 | 3.393 |
| ant-medium-v2 | 4456 | 4.393 | 997920 | 3.667 |
| pen-human-v1 | 4900 | 30.458 | 4950 | 31.609 |
| door-human-v1 | 9900 | 14.722 | 6679 | 2.927 |
| hammer-human-v1 | 9900 | 62.082 | 11260 | 6.756 |
| relocate-human-v1 | 9900 | 20.705 | 9892 | 9.184 |
| pen-cloned-v1 | 4900 | 30.458 | 492397 | 25.017 |
| door-cloned-v1 | 9900 | 14.722 | 991225 | 1.312 |
| hammer-cloned-v1 | 9900 | 62.082 | 992662 | 2.821 |
| relocate-cloned-v1 | 9900 | 20.705 | 992210 | 4.573 |

### C.2  MOTIVATION EXPERIMENT (FIGURE 1)

The figure 1 demonstrate a case to verify our motivation. Here we provide the details of its experimental setting. The illustration in Figure 1 is conducted in the maze2d-medium environment with dense reward. Expert data contains 5 trajectories, each with 600 transitions generated from the PD controller of D4RL Fu et al. (2020). As we claimed in Appendix C.1, the random data contains 5000 trajectories with 200 transitions each. Following the previous offline imitation learning, the auxiliary data is formed by mixing low-quality trajectories (5000 trajectories) and some

expert trajectories (ranging from 5 to 0 trajectories in Figure 1). The averaged results of DWBC (the state-of-the-art method in offline imitation learning) over three seeds are reported in Figure 1.

## C.3 COMPETING BASELINES.

We compare BCDP with the well-validated baselines and state-of-the-art offline imitation learning methods, includes:

- **BC-exp**: Behavior cloning on expert data $D^E$. The $D^E$ contains high-quality demonstrations but with limited quantity, and thus causes serious generalization problem.

- **BC-all**: Behavior cloning on the union of expert data $D^E$ and offline data $D^O$. BC-all is expected to be better than BC-exp as it integrates more offline data. However, when the quality of the offline data is low, BC-all may be weaker than BC-exp.

- **DemoDICE** (Kim et al., 2022): DemoDICE approximates the state-action distribution $d^\pi(s,a)$ to both expert data with offline data, treating offline data as a regularization, i.e., $\min_\pi KL(d^\pi||D^E) + \alpha KL(d^\pi||d^o)$ with the expectation of further improving performance based on expert data.

- **DWBC** (Xu et al., 2022a): DWBC regards the offline data as a mixture of expert-similar trajectories and low-quality trajectories, and apply positive-unlabeled learning to build a discriminator. The discriminator will evaluate unlabeled trajectories and provide an expert-similarity score, followed by a weighted behavior cloning.

- **OTIL** (Luo et al., 2023): OTIL uses optimal transport to label the rewards of offline trajectories based on their Wasserstein distance from expert trajectories. It then employs offline reinforcement learning algorithms to train an agent on the labeled dataset. We implement offline RL using IQL (Kostrikov et al., 2022), as described in the original paper.

- **UDS** (Yu et al., 2022): UDS labels all rewards from the unlabeled datasets with 0 (minimum rewards), and uses offline reinforcement learning algorithms to train the agent on the merged dataset. This method has been found to be effective for the utilization of low-quality data. Compared to their setting has a high-quality labeled dataset, our expert dataset does not have a ground-truth reward label. Instead, we label them with 1 (maximum rewards) to implement offline RL. Regarding the specific choice of offline RL algorithm, we have chosen our most similar offline RL algorithm, TD3+BC.

## C.4 MODEL ARCHITECTURE AND HYPER-PARAMETERS

For fair comparison, we follow the DWBC (Xu et al., 2022a) to build the actor network for all methods. In our BCDP, we follow the design of TD3+BC (Fujimoto & Gu, 2021) and build the critic network. For DemoDICE (Kim et al., 2022), we directly use their TensorFlow implementation in the experiments. All other methods are implemented via Pytorch.

Balance factor $\alpha$: Following the TD3BC paper, which also considers the trade-off between Q-learning and behavioral cloning, we consider balancing the optimization process by using the averaged batch-wise loss as a reference. Specifically, we define the batch-wise BC loss as $\lambda_1$ and the batch-wise Q loss as $\lambda_2$, then $\alpha = \lambda_2/\lambda_1$. This balance parameter will stabilize the absolute range of both values, similar to what was designed in TD3BC. The learning rates are roughly selected in 1e-3, 1e-4, and 1e-5. Specifically, maze2d-umaze, medium, and large use learning rates of 1e-3, 1e-4, and 1e-5 respectively, where more challenging tasks require smaller learning rates for a stable learning process. Mujoco and Adroit use a learning rate of 1e-4 and 1e-5 respectively. The frequency $t_{freq}$ of updating with policy gradient from the critic. We did not put much effort into adjusting $t_{freq}$ to improve performance. For Navigation, Locomotion, and Manipulation, the frequency $t_{freq}$ is set to 2, 3, and 3 respectively.

## C.5 SUPPLEMENTAL RESULTS UNDER VARYING BUDGETS

Figure 5 provides the results of the experiment in subsection 4.2 under all settings, and we can find that our method BCDP has achieved competitive results across varying expert budegts.

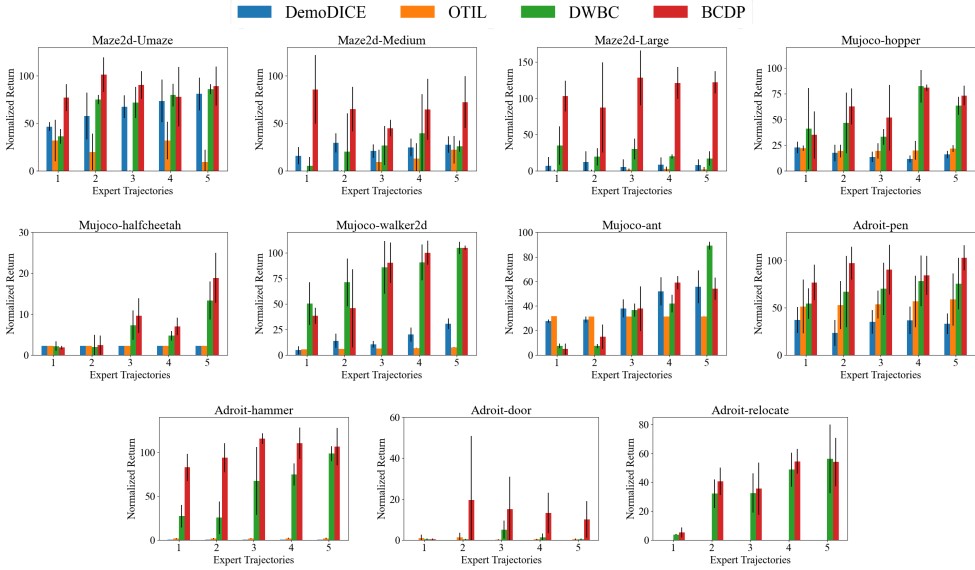

Figure 5: Comparision under varying number of expert trajectories.

## C.6 Scalability with Optimized Rewards

In this paper, we mainly focus on whether it is possible to remove previous assumptions on high-quality behavioral data and attempt to take advantage of the benefits of low-quality data. When the data contains some high-quality behavioral data, our method tends to be conservative, so its performance improvement is not as significant as in the setting with purely low-quality data. In such cases, we can also combine previous research to identify potential expert-similar data to further enhance performance. Here, we provide the corresponding experimental results. Specifically, we use DWBC and OTIL as reward labels for offline auxiliary data and then apply the BCDP method.

Table 3: Results under Optimized Rewards

|  | BCDP | DWBC-BCDP | OTIL-BCDP |
| --- | --- | --- | --- |
| maze2d-sparse-umaze-v1 | $132. \pm 22.0$ | $128. \pm 12.9$ | $\textbf{137.} \pm \textbf{6.59}$ |
| maze2d-sparse-medium-v1 | $137. \pm 11.4$ | $138. \pm 19.0$ | $\textbf{139.} \pm \textbf{9.84}$ |
| maze2d-sparse-large-v1 | $124. \pm 22.0$ | $111. \pm 14.8$ | $\textbf{137.} \pm \textbf{19.8}$ |
| hopper-medium-v2 | $\textbf{98.7} \pm \textbf{5.81}$ | $88.3 \pm 8.85$ | $59.4 \pm 0.97$ |
| halfcheetah-medium-v2 | $18.4 \pm 10.9$ | $15.6 \pm 11.1$ | $\textbf{40.5} \pm \textbf{3.79}$ |
| walker2d-medium-v2 | $\textbf{98.0} \pm \textbf{1.94}$ | $97.2 \pm 9.88$ | $96.1 \pm 4.79$ |
| ant-medium-v2 | $59.7 \pm 16.2$ | $14.1 \pm 13.8$ | $\textbf{94.8} \pm \textbf{4.07}$ |

The results of BCDP with optimized rewards demonstrate its potential when combined with previous reward annotation and trajectory recognition methods. Especially when the auxiliary data contains some relatively high-quality data, the performance of the BCDP framework can be further improved. It clearly supports the advantages of our proposal on scalability.

## C.7 Scalability with Different Reinforcement Learning methods

In this paper, we formulate the utilization of low-quality behavior data as an expert-state-distribution maximization problem, and address it via TD3. Here, TD3 can be replaced with different (offline) reinforcement learning methods. In this case, we conduct the experiments with popular methods, such as IQL (Kostrikov et al., 2022), CQL (Kumar et al., 2020), and BEAR (Kumar et al., 2019). We also provide the UDS with different RL implementations as we did in the main evaluation. For

the specific implementation of IQL, CQL, and BEAR, we follow the CORL benchmark [1] and the official code of BEAR [2]. The experimental results on maze2d are reported in the Table 4.

Table 4: Results with different offline reinforcement learning methods.

| | UDS (Ablation) | | | | BCDP (Ours) | | | |
|---|---|---|---|---|---|---|---|---|
| | TD3+BC | IQL | CQL | BEAR | TD3 | IQL | CQL | BEAR |
| sparse-umaze-v1 | 91.1 ± 22.9 | 49.0 ± 8.88 | 83.7 ± 86.3 | 21.9 ± 6.60 | **132.** ± **22.0** | 111. ± 36.6 | 103. ± 18.4 | 94.5 ± 47.8 |
| sparse-medium-v1 | 97.0 ± 20.0 | 90.4 ± 29.9 | 67.3 ± 63.2 | -1.5 ± 5.04 | **137.** ± **11.4** | 125. ± 20.1 | 83.3 ± 38.2 | 87.2 ± 62.4 |
| sparse-large-v1 | 134. ± 26.0 | 100. ± 17.5 | 164. ± 37.5 | 0.67 ± 5.50 | 124. ± 22.0 | **206.** ± **14.9** | 105. ± 13.8 | 16.0 ± 16.1 |
| hopper-random-v2 | 1.15 ± 0.46 | 5.76 ± 1.99 | 32.0 ± 2.64 | 4.86 ± 2.47 | **73.2** ± **9.66** | 46.5 ± 29.8 | 59.0 ± 9.15 | 4.86 ± 2.47 |
| halfcheetah-random-v2 | 4.62 ± 1.15 | 2.25 ± 0.00 | 6.67 ± 1.63 | 2.25 ± 0.00 | **18.8** ± **6.11** | 3.27 ± 2.60 | 1.82 ± 0.52 | 3.54 ± 2.31 |
| walker2d-random-v2 | -.11 ± 0.00 | 45.3 ± 48.6 | 7.41 ± 0.60 | 0.90 ± 0.01 | **105.** ± **2.08** | 88.1 ± 32.7 | 96.1 ± 9.33 | 104. ± 3.72 |
| ant-random-v2 | 30.4 ± 2.99 | **71.0** ± **0.76** | 39.8 ± 6.35 | 30.9 ± 0.02 | 54.1 ± 9.01 | 62.3 ± 12.9 | 47.0 ± 20.1 | 58.6 ± 4.30 |

From the results, we could find that our BCDP has scalability for different offline reinforcement learning methods. Compared to directly applying offline reinforcement learning methods (UDS), BCDP brings consistent performance improvement. In addition, we can also see that advanced offline RL algorithms, such as IQL, can help BCDP achieve further performance growth in some cases. However, overall, TD3 remains a stable choice, which is also simple and effective.

## C.8 CONSERVATIVE ANALYSIS OF OFFLINE RL METHODS

In section 3.2, we point out that existing offline RL methods are difficult to apply to offline imitation learning problems. This is because, without reward labels to guide optimization, their conservative regularization may mislead policy learning towards low-quality behaviors, which are prevalent in auxiliary data. Here, we provide further experimental evidence to support this point.

Table 5: Hyper-parameters of conservative regularization.

| BEAR | | | | | | |
|---|---|---|---|---|---|---|
| | UDS (Ablation) | | | BCDP (Ours) | | |
| weighted factor | 10 | 3 (selected) | 0.1 | 10 | 3 | 0.1 (selected) |
| sparse-umaze-v1 | 18.7 ± 5.18 | 21.9 ± 6.60 | -2.89 ± 12.4 | 57.1 ± 15.7 | 57.5 ± 29.1 | 94.5 ± 47.8 |
| sparse-medium-v1 | -1.7 ± 5.57 | -1.5 ± 5.04 | -1.91 ± 5.30 | 7.97 ± 12.3 | 32.0 ± 13.4 | 87.2 ± 62.4 |
| sparse-large-v1 | 0.63 ± 5.43 | 0.67 ± 5.50 | 0.54 ± 5.28 | 7.44 ± 17.1 | 7.69 ± 16.3 | 16.0 ± 16.1 |
| CQL | | | | | | |
| | UDS (Ablation) | | | BCDP (Ours) | | |
| weighted factor | 10 | 1 | 0.1 (selected) | 10 | 1 | 0.1 (selected) |
| sparse-umaze-v1 | -9.11 ± 11.1 | 18.7 ± 61.2 | 83.7 ± 86.3 | 38.8 ± 23.3 | 20.0 ± 11.0 | 103. ± 18.4 |
| sparse-medium-v1 | -4.42 ± 0.48 | -4.71 ± 0.45 | 67.3 ± 63.2 | 8.93 ± 11.4 | 6.00 ± 12.1 | 83.3 ± 38.2 |
| sparse-large-v1 | -1.61 ± 0.79 | -2.17 ± 0.59 | 164. ± 37.5 | 12.0 ± 12.6 | 17.1 ± 17.3 | 105. ± 14.8 |

CQL Kumar et al. (2020) and BEAR Kumar et al. (2019) reduce out-of-distribution error by maximizing the lower bound of the Q-function and minimizing the MMD distance between the policy and the behavior data distribution, respectively. We validate them in the offline imitation learning problem and adjust their weighted factor of conservative regularization term for observation. The weight candidates of CQL and BEAR are suggested from the CORL and Zhang et al. (2021).

From the results, we found that, for the original offline RL methods, smaller conservative regularization leads to higher returns, but also higher variance (such as CQL). This supports the discussion in our paper that the conservative terms in existing offline RL methods may be too conservative, making it difficult to handle situations with a large amount of low-quality data, such as offline IL. When facing the offline IL problem, a more appropriate choice may be to remove the regularization term, such as directly using the TD3 algorithm like us, or consider advanced offline RL methods, such as IQL. Both of these options have demonstrated higher and more stable performance in Table 4 above. Additionally, after reducing the weight of the conservative regularization term, the BCDP framework we proposed can also help CQL and BEAR achieve performance improvement, demonstrating its scalability.

---

[1] https://github.com/tinkoff-ai/CORL/tree/main
[2] https://github.com/ryanxhr/BEAR

## C.9 LEARNING CURVES OF MAIN RESULTS

Here we provide the corresponding learning curves of our main experiments in the Table 1.

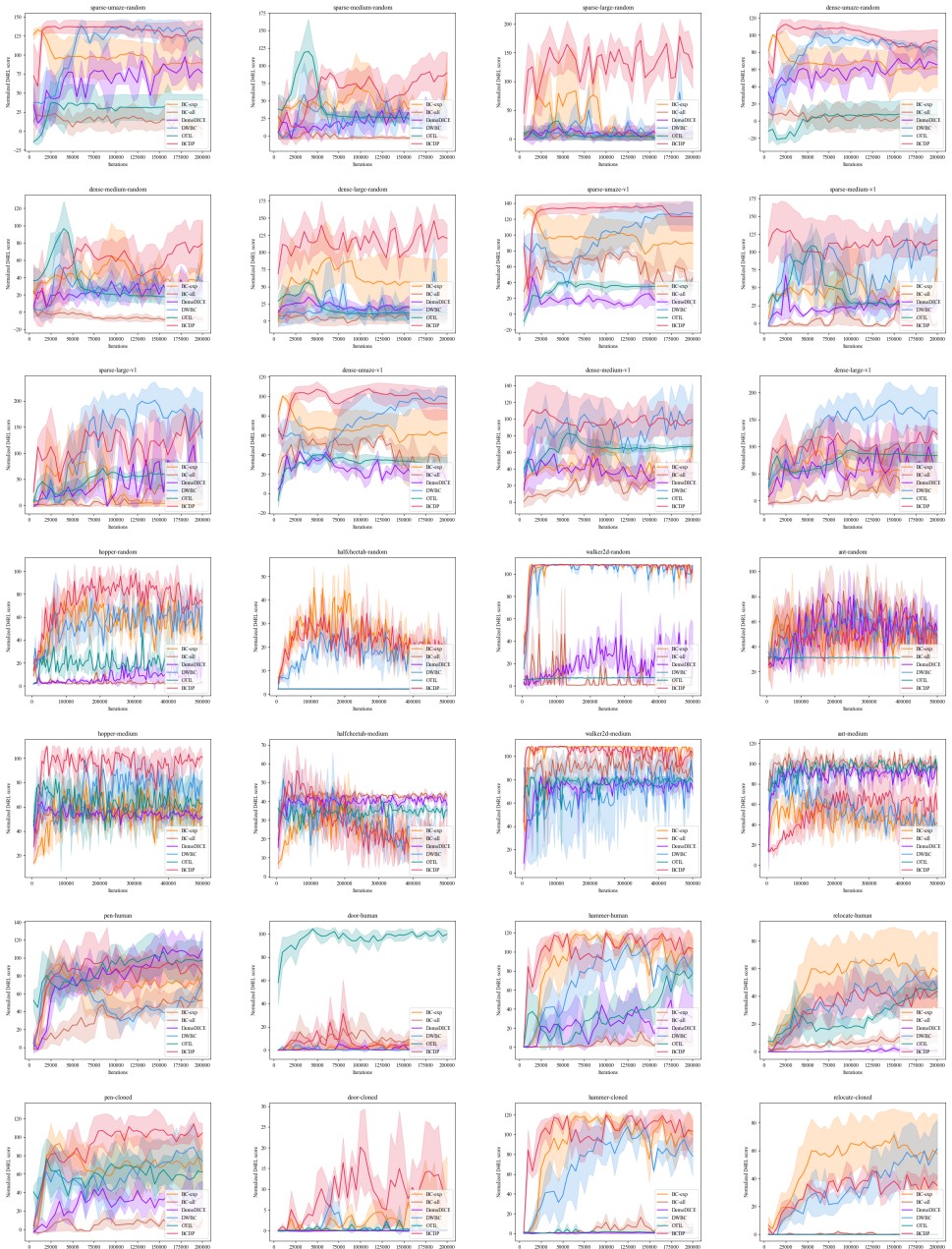

Figure 6: Learning curves of the main results.

As shown in Figure 6, in most cases, especially when the quality of the auxiliary data is low, our BCDP achieves consistent and stable performance improvement.

