# OpenReview forum: "Offline Imitation Learning without Auxiliary High-quality Behavior Data"
_ICLR.cc/2024/Conference — Submitted to ICLR 2024_

### Official Review · Reviewer_uGnS · 2023-10-15

**Soundness:** 3 good
**Presentation:** 3 good
**Contribution:** 3 good
**Rating:** 6
**Confidence:** 3

**Summary:**

This paper proposes BCDP, a method for offline imitation learning with low-quality behavior data from which good policy cannot be directly extracted. The idea is simple: mimic expert on expert state, and reach expert state as frequently as possible when the agent is off the expert trajectory. Inspired by TD3+BC, BCDP designs an auxiliary reward that is 1 on expert state and 0 on non-expert state, and maximize a weighted sum of log likelihood on the expert dataset and auxiliary reward over the union of expert and non-expert dataset. On several testbeds with low-quality (e.g. random) auxiliary dataset, BCDP outperforms many offline imitation learning methods.

**Strengths:**

1. The paper answers a timely and interesting question of today's offline Imitation Learning (IL) community on how to utilize low-quality auxiliary data, which is not well-addressed by prior work and useful in real-life applications.

2. The idea proposed by the paper is sound and simple, and is backed up by theoretical lower bound of performance and experiment results with ablation.

**Weaknesses:**

**1. The reward design seems to be too strict.** Currently, judging from section 3.2, the reward for a state-action pair is 1 if and only if the state-action pair belongs to the expert dataset. However, with continuous state/action space and low quality auxiliary dataset, it is very likely that none of the state-action pairs in $D^O$ has a reward of 1 - and with pertubation in the continuous environment, it is even likely that every state appear in $D^O\cup D^E$ is unique and no state in the auxiliary dataset $D^O$ lead to any reward. In such case, the optimization relies on the smoothing effect of the critic network. Thus, I suggest to add an ablation on replacing the current, identical reward with the discriminator used in prior work such as DWBC, ORIL or DemoDICE.

**2. The hyperparameter is not specified, and some auxiliary plots are missing.** Currently, the value of some important hyperparameters are missing, e.g., weight for maximizing Q in the objective, network architecture, learning rate, number of steps to update, frequency of updating with policy gradient from critic. Also, the paper does not include learning curves.

3. I encourage the authors to include discussion on the **limitation of the current work.**



Other Minor errors:

1.Typos:

a) in "background and formulation", "... which is much cheap to obtain" should be "... which is much **cheaper** to obtain";

b) in the figure 2, "How to achieve a expert-observed state" should be "How to achieve **an** expert-observed state";

c) in Algorithm 1, "calcualte" should be "calculate", "datset" should be "dataset";

d) In the second line of section 4, "relatuve" should be "relative".

2. Other notation issues:

a) d is used as both the policy gradient update frequency and state distribution (occupancy); I suggest to adjust the notations to prevent misunderstanding.

b) the notation $\mathbb{I}$ is not formally introduced, which is a crucial factor to the reward design.

**Questions:**

I have two question apart from those listed in the weakness:

1. How does the method perform compared to model-based imitation learning method (e.g., [1, 2, 3])? It might be hard to extract useful policy from low-quality behavior data, but those data could be useful for building up dynamic models, and thus help model-based methods.

2. Why is policy gradient with respect to Q function updated only every d steps? intuitively, updating it every step with a smaller coefficient would make the gradient descent process smoother.

**References**

[1] W. Zhang et al. Discriminator-Guided Model-Based Offline Imitation Learning. In CoRL, 2022.

[2] A. Hu et al. Model-Based Imitation Learning for Urban Driving. In NeurIPS, 2022.

[3] R. Kidambi et al. MobILE: Model-Based Imitation Learning From Observation Alone. In NeurIPS, 2021.

---

> ### Author Response · Authors · 2023-11-15
>
> Thanks for your positive comments and valuable suggestions. Below are our responses to the concerns.
>
> **W1: The reward design seems to be too strict**
>
> We obtained this reward design by maximizing the expert-state distribution: samples in the expert dataset have a reward of 1, and samples in the auxiliary dataset have a reward of 0. This design may be conservative because we mainly focus on the utilization of low-quality data, thus only utilizing transition information, which is independent of behavior quality. When the data contains some expert-similar trajectories, we can also replace the current identical reward with the discriminator used in prior work to make use of it.
> Here we provide the corresponding additional experimental results. Specifically, we use DWBC, and OTIL to annotate the reward labels for offline auxiliary data, and then apply the BCDP method with optimized rewards.
>
> |   | Strict-BCDP | DWBC-BCDP | OTIL-BCDP |
> | --- | --- | --- | --- |
> | maze2d-sparse-umaze-v1 | 132.+/-22.0 | 128.+/-12.9 | **137.+/-6.59** |
> | maze2d-sparse-medium-v1 | 137.+/-11.4 | 138.+/-19.0 | **139.+/-9.84** |
> | maze2d-sparse-large-v1 | 124.+/-22.0 | 111.+/-14.8 | **137.+/-19.8** |
> | hopper-medium | **98.7+/-5.81** | 88.3+/-8.85 | 59.4+/-0.97 |
> | halfcheetah-medium | 18.4+/-10.9 | 15.6+/-11.1 | **40.5+/-3.79** |
> | walker2d-medium | **98.0+/-1.94** | 97.2+/-9.88 | 96.1+/-4.79 |
> | ant-medium | 59.7+/-16.2 | 14.1+/-13.8 | **94.8+/-4.07** |
>
>
> The results of BCDP with optimized rewards demonstrate its potential when combined with previous reward annotation methods. Especially when the auxiliary data contains some relatively high-quality data, the performance of the BCDP framework can be further improved. It clearly supports the advantages of our proposal on scalability. We have added them to Appendix C.6 as additional supplementary material.
>
> **W2: The detailed hyperparameters and some auxiliary plots**
>
> We have provided the detailed hyper-parameters and the learning curves in Appendix C.4 and C.9 respectively.
>
> **W3: Limitation of the current work.**
>
> In this work, our main focus is to remove the traditional assumption of behavior quality in auxiliary data and leverage the transition information to assist in imitation learning. It is worth noting that our approach may be conservative, particularly when high-quality data is available in the dataset. However, as you suggested, we can further enhance the performance of BCDP by combining previous reward annotation methods. We acknowledge that our exploration in this area is currently limited, we will combine these two different approaches of utilizing auxiliary data in future research to achieve a better balance between performance and robustness. Moreover, this study currently only considers the model-free solution. Exploring how to integrate existing model-based techniques to guide the agent toward expert-observed states is also a promising direction that we will attempt in future work.
>
>
> **W4: Typos and notation issues**
>
> Thanks for your suggestions, we have revised them in the new version.

---

> ### Author Response · Authors · 2023-11-15
>
> **Q1: Discussion with model-based methods**
>
> Thanks for your valuable suggestions. When conducting the experimental comparison, we mainly referred to previous offline imitation learning methods and their competing baselines [1-3]. **Our proposal is also a model-free solution, so we primarily compare it with model-free methods, which are widely adopted in this topic**. Comparing with model-based methods may not be fair as they require additional training on dynamics models.
>
> [1] Demodice: Offline imitation learning with supplementary imperfect demonstrations. ICLR 2022
> [2] Discriminator-weighted offline imitation learning from suboptimal demonstrations. ICML 2022
> [3] Optimal transport for offline imitation learning. ICLR 2023
>
> It is true that model-based methods also utilize transition information, (Reviewer Xzkb also pointed out it). Therefore, we add the discussion with the model-based methods in Appendix B to **clarify the difference between our proposal and the model-based methods**. Overall, **our approach is fundamentally different from existing model-based methods in terms of utilizing transition information**. Existing model-based methods [4-7] may still struggle to handle the problem we propose.
>
> 1. On the one hand, since the data does not have reward labels, the trajectories sampled by model-based methods are still unlabeled. We still need to consider how to utilize these low-quality behavior data that have been sampled. In other words, the problem we presented in this paper is still unsolved.
> 2. On the other hand, these methods [4,7] still perform distribution matching between the generated trajectories and the state-action distribution of expert behavior ($p(s, a)$) to annotate reward labels, which is consistent with our summary of existing work in equations 2 and 3. As we discussed in sections 2.2 and 2.3, they cannot provide optimization objectives on expert-unobserved states because this similarity can only rely on the model's function approximation ability to distinguish the quality of unlabeled trajectories and utilize them. This makes it difficult for them to effectively utilize offline data, which may contain many expert-unobserved states.
>
> On the contrary, when utilizing transition information, we consider the subsequent expert-state distribution maximization. This approach does not expect the sampled trajectories to necessarily contain samples with a similar state-action distribution to expert data. Instead, it provides guidance to the agent towards expert-observed states through the transition information between states, thereby improving the lower bound of long-horizon return.
>
> [4] Discriminator-Guided Model-Based Offline Imitation Learning. CoRL, 2022.
> [5] Model-Based Imitation Learning for Urban Driving. NeurIPS, 2022.
> [6] MobILE: Model-Based Imitation Learning From Observation Alone. NeurIPS, 2021.
> [7] Mitigating Covariate Shift in Imitation Learning via Offline Data with Partial Coverage. NeurIPS’21
>
> **Q2: Update of Q function**
>
> The delayed update is designed in the TD3 algorithm, aiming to decrease the error per update. In simple terms, the delayed update allows the critic to stabilize before updating the actor, avoiding the critic's changing too much and causing the actor's optimization direction to become unstable. For detailed information and further discussion, please refer to the TD3 paper [8].
>
> [8] Addressing Function Approximation Error in Actor-Critic Methods. ICML 2018

---

> > ### Comment · Reviewer_uGnS · 2023-11-16
> > **Response to the Rebuttal**
> >
> > Thanks for the authors' detailed response. I think my questions are addressed; the phenomenon that the better design of the reward varies with the auxiliary data quality is interesting. However, as reviewer K5Tt pointed out, the method is somewhat similar to existing methods, such as TD3+BC. Overall, I decide to keep my current score for now.

---

### Official Review · Reviewer_K5Tt · 2023-10-30

**Soundness:** 2 fair
**Presentation:** 2 fair
**Contribution:** 2 fair
**Rating:** 5
**Confidence:** 4

**Summary:**

The paper studies the problem of Offline Imitation Learning (OIL) in the absence of auxiliary high quality data samples. Towards this goal, authors propose Behavior Cloning with Dynamic Programming (BCDP) which maximizes probability of transition to expert-observed states. BCDP abstracts data quality from IL by implementing BC on expert data samples and DP on unlabeled offline data samples. Practically, BCDP incorporates the TD3 algorithm wherein DP is carried out over value functions using the Bellman equation formulation. Empirical evaluation on a range of tasks demonstrates competitive performance to IL baselines.

**Strengths:**

* The paper is well organized.
* Experiments carried out by authors are sufficient.

**Weaknesses:**

* **Motivation:** While the paper aims to leverage low-quality data samples, its motivation for the same is unclear. Instead of highlighting the algorithmic design choices, authors focus on the split of datasets. This corresponds to implementing offline RL on only a subset of data samples. For instance, Section 1 and Figure 1 do not motivate the need for an expert-specific dataset split. Moreover, it is unclear as to what conclusions can be drawn from Figure 1. Authors should explain the task definition, complexity of task completion, number of trajectories, number of samples in each trajectory for the BC expert and the ratio of expert to non-expert data split.
* **Practical Implementation:** While the work combines offline IL with DP, it leads to the well-established and pre-existing paradigm of offline RL. Practical implementation of BCDP presented in Section 3.2 is akin to applying offline RL on a different dataset split. As the authors note, BCDP is a special case of TD3+BC. However, its differences from TD3+BC or other offline RL algorithms remain unclear.
* **Choice of Baselines:** While the authors state that their approach is similar to TD3+BC and offline RL, BCDP is not compared to these methods. Authors claim that "they have selected TD3+BC as our most similar offline RL algorithm, which allows it to be considered as an ablation study for our approach". However, an abaltion study of TD3+BC with BCDP is missing. It would be worthwile to compare BCDP with TD3+BC or a conservative algorithm such as CQL. This would help evaluate the utility of BCDP with established offline RL methods.
* **OOD Evaluation:** The requirement and intuition behind DRG metric is unclear. From my understanding, DRG is a state occupancy measure based on agent's visitation towards particular states. Hence, it does not quantify the policy's performance at test time (as per Q3). A positive DRG does not indicate how the agent performed on unobserved states. It only indicates that the agent successfully evaded unobserved states. This leaves Q3 unanswered. In order to evaluate OOD performance, authors could use standard methods/metrics. For instance, BCDP could be evaluated on a set of heldout states or initialized in a new state (or random seed). Similarly, authors could measure the confidence of agent by assessing the probability of actions taken in unobserved states.
* **Writing and Presentation:** In general, writing and presentation should be refined. Sentences and verbs could be made complete and grammatical errors could be reduced. Authors should also provide the missing ablations of TD3+BC and clearly explain what they wish to observe from Q3.

### Minors
* methods has achieved $\rightarrow$ methods have achieved
* data with uncertain $\rightarrow$ data of uncertain
* suboptimal polices $\rightarrow$ suboptimal policies
* which optimize the $\rightarrow$ which optimizes the
* learning provide a $\rightarrow$ learning provides a
* an action could $\rightarrow$ an action that could
* relatuve $\rightarrow$ relative

**Questions:**

* Can you please explain the task definition and setting of Figure 1? How many trajectories are present in the dataset? How many samples are present per trajectory? What is the ratio of expert to non-expert data samples?
* How is BCDP different from TD3+BC? What would be the potential advantages of using BCDP over TD3+BC with expert demonstrations? How does BCDP compare to TD3+BC?
* How does BCDP compar to existing offline RL methods such as CQL, BEAR, BRAC or IQL?
* What does DRG indicate? How does DRG quantify the performance of policy in unobserved states? Can BCDP be evaluated on a set of heldout states or new random seeds? Does the agent present high confidence in its actions on unobserved states?

---

> ### Author Response · Authors · 2023-11-15
>
> **W1 and Q1: Motivation and the details of Figure 1.**
>
> Thanks for your comments. However, it looks like you have some misconceptions about motivation. We clarify our motivation in the following:
>
> The most recent successes in offline imitation learning heavily rely on high-quality behavior auxiliary data. For example, previous studies either manually integrate expert data into sub-optimal data or introduce noise into expert data to construct the auxiliary unlabeled dataset [1-4]. Unfortunately, the high-quality behavior data assumption neither holds in real-world applications nor in the benchmark datasets (e.g., D4RL). Facing low-quality behavior data, existing methods suffer severe performance degradation (as shown in Figure 1). How to exploit the easily obtained low-quality behavior data is a realistic yet unaddressed problem. We are the first ones to attempt to remove the high-quality behavior auxiliary data assumption for offline imitation learning and make a successful step. We believe this sheds new light on the study of offline imitation learning.
>
> [1] Behavioral cloning from noisy demonstrations. ICLR 2021
> [2] Demodice: Offline imitation learning with supplementary imperfect demonstrations. ICLR 2022
> [3] Discriminator-weighted offline imitation learning from suboptimal demonstrations. ICML 2022
> [4] Imitation learning from imperfection: theoretical justifications and algorithms. NeurIPS 2023
>
> In Figure 1, we conducted a motivation evaluation on the maze2d task to observe the performance curve of the algorithm when we reduce the number of expert trajectories in the auxiliary data or even when it only contains low-quality behavior data (x-axis = 0). As a result, we found the performance of DWBC (a SOTA method) degrades significantly as the number of expert trajectories in the unlabeled data decreases. These findings further support the viewpoint that the previous methods may not effectively utilize low-quality data.
> The experimental details of Figure 1 have been added to Appendix C.2.
>
> **W2 and Q2: Comparison between BCDP and TD3+BC**
>
> In this work, we formalize the utilization of low-quality data in offline imitation learning as an **expert-state distribution maximization** problem and it can be instantiated using different dynamic programming methods. We chose TD3 because it is concise and stable. This does not mean that BCDP is a special case of TD3+BC. The difference arises from **two completely different purposes**. The behavioral cloning in the original TD3+BC is for conservatism, to keep the learned policy close to the offline policy and eliminate the estimation bias of TD3. Our goal with behavior cloning is to follow the imitation learning of expert data. Our TD3 (DP part) on the offline dataset ($ D^O \cup D^E $) is to enable the agent to transition from expert-unobserved states to expert-observed states. Compared to previous imitation learning methods, this provides a new direction for the safe utilization of low-quality data, that is, using their transition information to help imitation learning.
>
> **We actually compared it with the TD3+BC in all settings**. As we claimed in the raw paper: “UDS (Yu et al., 2022) labels all rewards from the unlabeled datasets with 0 and utilizes offline reinforcement learning algorithms to train the agent on the merged dataset. We have selected TD3+BC as our most similar offline RL algorithm, which allows it to be considered as an ablation study of our approach.” In all experiments, the UDS represents the raw TD3+BC in the offline imitation learning setting, as an ablation. In the offline imitation learning setting, where all data is unlabeled, applying offline RL algorithms on the union dataset is equivalent to UDS (Unlabeled Data Sharing), where expert data has reward label 1 and auxiliary data has reward label 0. As shown in the original paper's results and corresponding discussions, TD3+BC performs unsatisfactorily when the behavior quality of offline data is low.

---

> ### Author Response · Authors · 2023-11-15
>
> **W3 and Q3: Comparision with more offline RL methods**
>
> Thank you for your interest in comparing BCDP with more offline RL methods. We provide here a comparison with some popular offline RL methods, including CQL, IQL, and BEAR. Furthermore, we can also implement different (offline) RL methods in the BCDP framework, as an alternative to TD3.
> |  | UDS(Ablation) |  |  |  | BCDP (Ours) |  |  |  |
> | --- | --- | --- | --- | --- | --- | --- | --- | --- |
> |  | TD3+BC | IQL | CQL | BEAR | TD3 | IQL | CQL | BEAR |
> | sparse-umaze-v1 | 91.1 ± 22.9 | 49.0 ± 8.88 | 83.7 ± 86.3 | 21.9 ± 6.60 | **132. ± 22.0** | 111. ± 36.6  | 103.± 18.4 |  94.5 ± 47.8 |
> | sparse-medium-v1 | 97.0 ± 20.0 | 90.4 ± 29.9 | 67.3 ± 63.2 |  -1.5 ± 5.04 | **137. ± 11.4**  | 125. ± 20.1 | 83.3 ± 38.2 | 87.2 ± 62.4 |
> | sparse-large-v1 | 134. ± 26.0  | 100. ± 17.5  | 164. ± 37.5 |  0.67 ± 5.50  | 124. ± 22.0  | **206. ± 14.9**  | 105. ± 13.8 | 16.0 ± 16.1 |
> | hopper-random-v2 | 1.15 ± 0.46 | 5.76 ± 1.99 |  32.0 ± 2.64 |  4.86 ± 2.47 | **73.2 ± 9.66**  | 46.5 ± 29.8 | 59.0 ± 9.15  | 4.86 ± 2.47 |
> | halfcheetah-random-v2 | 4.62 ± 1.15 | 2.25 ± 0.00  |  6.67 ± 1.63 |  2.25 ± 0.00 | **18.8 ± 6.11** | 3.27 ± 2.60  |  1.82 ± 0.52 |  3.54 ± 2.31 |
> | walker2d-random-v2 | -.11 ± 0.00 | 45.3 ± 48.6 | 7.41 ± 0.60 | 0.90 ± 0.01 | **105.± 2.08** | 88.1 ± 32.7 |  96.1 ± 9.33 | 104. ± 3.72 |
> | ant-random-v2  | 30.4 ± 2.99 | **71.0 ± 0.76** | 39.8 ± 6.35 | 30.9 ± 0.02 | 54.1 ± 9.01 | 62.3 ± 12.9 |  47.0 ± 20.1 | 58.6 ± 4.30 |
>
> From the results, we could find that our BCDP has scalability for different offline reinforcement learning methods. Compared to directly applying offline reinforcement learning methods (UDS), BCDP brings consistent performance improvement. These results further demonstrate the scalability of our BCDP. We have added these results to  Appendix C.7.
>
> **W4 and Q4: OOD Evaluation**
>
> We answered Q3 (Q3 in the main paper) in two parts: 1) Whether the learned agent tends to move towards expert-observed states when it is in expert-unobserved states, along with the optimization objectives we provided. 2) Whether BCDP improves the agent's long-term returns on expert-unobserved states. **We addressed them in Figure 4(a) and (b)**.
>
> (1) DRG serves as a characterization of an agent's transition from one state $s$ to another state $s’$, with respect to the 1-NN distance from expert observed states, as defined by its definition. Specifically, $min_{s_1\in D^E} \lVert s-s_1 \rVert_2$ represents the 1-NN distance from state s to the expert dataset $D^E$ and $min_{s_2\in D^E} \lVert s’-s_2 \rVert_2$ is similar, where $s’$ represents the expected next state given policy $\pi$. The evaluation is performed on navigation in the maze2d. Therefore, the Euclidean distance can serve as an approximate measure for state distances. If DRG(s) > 0, it means that the agent will transition from state $s$ to a state $s'$ that is closer to the expert-observed states. Figure 4(a) illustrates that the BCDP actually has a positive expected DRG and tends to the expert-observed states, particularly in states with larger out-of-distribution (OOD) distances (indicated by a bigger value on the x-axis).
>
> (2) As you pointed out, DRG does not directly quantify the performance of policy in unobserved states. Therefore, we also presented the long-term return in Figure 4(b). The Long-Term-Return ($s$) represents the sum of the agent's subsequent 400 steps starting from state $s$, calculated through 1000 random repetitions. We report the Long-Term-Return($s$) in Figure 4(b). Please note that Figure 4(b) is aligned with Figure 4(a), indicating that the positive DRG actually brings higher long-term returns for our agent. These results support that we have implemented our proposal, which is to leverage transition information from low-quality behavior data and improve the long-term return on expert-unobserved states, aligning with the theoretical motivation.
>
> **W5: Writing and Presentation**
> Thanks for your kind suggestions. We have revised them in the new version.

---

> > ### Comment · Reviewer_K5Tt · 2023-11-16
> > **Response to Authors**
> >
> > I thank the authors for providing a detailed response to reviews. After going through the responses of authors and other reviews, my following concerns remain-
> >
> > * **Motivation:** Authors explained the decrease in return when number of expert trajectories are reduced. However, my concern is centered at the reasoning behind this decrease. Irrespective of the choice of method, returns will always decrease if the number of expert trajectories in a dataset are reduced. This is true for standard imitation learning methods as well as offline RL methods. DWBC being a state of the art method does not alter this general result. It would be helpful if authors could evaluate BCDP for varying expert trajectories under the same experiment setting.
> > * **Practical Implementation:** Authors mention that BCDP abstracts imitation from dynamic programming for conservatism and in-distribution transitions respectively. Practically, BCDP still serves the same purpose as Offline RL methods. While IL is used to retrieve the data distribution, DP is used to stitch together trajectories which constrain the agent to transition to expert-observed states. Note that this is the primary focus of offline RL methods utilizing temporal difference learning. Additionally, authors also highlight that BCDP enables safe utilization of data using transition information. However, this information is solely aggregated and learned by the value function which makes DP restrictive to the setting of off-policy policy improvement. It still remains unclear as to what differences or benefits BCDP offers over offline DP/TD methods in the sub-optimal data regime.

---

> ### Author Response · Authors · 2023-11-17
>
> Thank you for your reply and for giving us the opportunity to clarify some misconceptions.
>
> **Response to motivation**
>
> In offline imitation learning with auxiliary data, there are expert dataset $D^E$ and auxiliary offline dataset $D^O$. Indeed, for general offline imitation learning methods, the return decreases if the number of expert trajectories in $D^O$ is reduced. However, a notable fact in Figure 1 is that the **performance of DWBC (which works on $D^E \cup D^O$) degrades to being weaker than BC on $D^E$**. This contradicts the goal in this field of utilizing auxiliary data as an enhancement to $D^E$.
>
> Thanks for your interest in our BCDP. We have provided the corresponding results under the same settings as shown below.
>
>
> | Expert trajectories mixed in $D^O$ | 5 | 4 | 3 | 2 | 1 | 0 |
> | --- | --: | --- | --- | --- | --- | --- |
> | BC (baseline on $D^E$) | | | | | | 37.6 ± 19.2   |
> | DWBC  | 59.5 ± 17.3 | 56.2 ± 32.3 | 54.8 ± 4.22 | 40.4 ± 18.9 | 30.0 ± 14.0$\downarrow $ | 25.8 ± 5.78 $\downarrow $ |
> | BCDP | 90.3 ± 9.68 | 87.9 ± 21.4 | 84.0 ± 21.0 | 82.9 ± 31.4 | 75.3 ± 29.5 | 72.4 ± 27.2 |
>
>
> From the results, we can find that **BCDP can efficiently utilize auxiliary data $D^O$ from low-quality behavioral policy** and consistently outperform BC, **achieving the goal of using an offline auxiliary dataset to assist limited expert data $D^E$**. Additionally, we can observe that BCDP achieved a performance of 72.4 in $D^O$ even without expert trajectories, surpassing the performance of DWBC with a mixture of expert trajectories in auxiliary data. This highlights the potential of low-quality behavior data, which was overlooked in previous OIL research, and may pave the way for future advancements in the field (as reviewer Yds8 said).
>
> **Response to Practical Implementation**
>
> Regarding the comment mentioned, "Authors mention that BCDP abstracts imitation from dynamic programming for conservatism and in-distribution transitions respectively," we would like to clarify that this statement is not accurate.
> As presented in section 3, we follow the classic imitation learning approach by using BC to imitate the expert data $D^E$. When **BC guarantees the return on expert-observed states** (as claimed in our lemma 1), we further point out that the transition information in the auxiliary data is independent of the behavior quality, even if it comes from random exploration. Therefore, we propose the objective of maximizing expert-state distribution and implementing it through DP / TD algorithms. This is a **completely different purpose from offline RL**, such as TD3+BC, where BC is used to keep the learned policy close to the offline policy and eliminate the estimation bias of TD3.
>
> Let's consider a specific scenario as an example (Figure 2). In the navigation task, we have an expert trajectory ($D^E$) that can reach the goal and a large amount of random exploration on the map as auxiliary data ($D^O$), which means uniformly random actions on many states. **When applying TD3+BC on the union dataset, its BC term limits the agent to learn a uniformly random action and then fail.** In contrast, our BC only applies to expert data $D^E$, ensuring the ability in expert-observed states. TD3 maximizes expert-state distribution and enables the agent to transition from expert-unobserved states to expert-observed states (as shown in Figure 2).
>
> We have already pointed out this in Section 3.2, which makes TD3+BC difficult to adapt to low-quality auxiliary data. Extending to other Offline RL algorithms, **their conservative regularization limits their effectiveness in offline imitation learning problems**. The **comprehensive experimental results also supported this point**. The corresponding explanations are provided in Section 4.1 and Appendix C.7 and C.8.
>
> Overall, our work bridges the offline IL and offline RL, providing a guiding framework for introducing existing mature RL algorithms in offline IL. More importantly, it brings a new perspective to the existing offline IL, which mainly focuses on the utilization of expert-similar samples, showing that low-quality behavioral data could also help with imitation learning.

---

> > ### Comment · Reviewer_K5Tt · 2023-11-19
> > **Response to Follow Up**
> >
> > I thank the authors for responding to my concerns. When compared to DWBC, BCDP does leverage low-quality data effectively. In the absence of expert trajectories, BCDP still presents higher returns. Regarding the practical implementation, BC and DP counterparts play analogous roles of policy evaluation and policy improvement as in Offline RL. While similar to TD3+BC in many respects, BCDP leverages its expert and auxilary data distributions in a different way. I am not sure whether this merits as a significant contribution for the offline IL and RL communities but empirical evidence suggests potential for impact. Regarding empirical evaluation, authors demonstrate that BCDP is on par with conservative offline RL algorithms while leveraging low-quality auxilary data. Based on the empirical evaluation and the improved quality of the paper, I would like to adjust my score. I thank the authors for their efforts.

---

> > > ### Author Response · Authors · 2023-11-20
> > >
> > > Thank you very much for the re-evaluation! We are glad to hear that our response solved your concerns. Once again, we sincerely appreciate your time and valuable suggestions, which help us improve this work and make it more solid.

---

### Official Review · Reviewer_Xzkb · 2023-10-31

**Soundness:** 3 good
**Presentation:** 4 excellent
**Contribution:** 3 good
**Rating:** 8
**Confidence:** 4

**Summary:**

This paper proposes an effective use of low-quality offline data for off-policy imitation learning. They propose BCDP, which essentially does BC on the expert data while minimizing a SQIL-like Q-learning loss on both the expert data and offline data. This is similar to TD3+BC, where BC is used to provide some form of closeness to the offline data while TD3 maximizes an RL objective when rewards are present in the dataset. Experiments across multiple domains in the D4RL benchmark suite shows that BCDP outperforms other model-free offline IL baselines.

**Strengths:**

First of all, the paper's performance curves are very solid, especially for long-term return environments where the sparse reward used by BCDP could hamper learning. The experimental setup is pretty solid and covers all bases across the D4RL benchmark, with notable good results in sparse reward domains and domains with very low-quality offline data, such as datasets collected by random agents.

The paper was also well written and easy to follow. The graphs were somewhat unusual to see on an RL paper, but relevant in the context of what the paper is trying to show. It was easy to understand the author's reasoning throughout the paper.

**Weaknesses:**

There are some missing citations I think: for example, there has been some work on the model-based side with offline imitation learning, but with the assumption that the offline data has coverage over the expert traces in the state space [1]. This is reminiscent of what this paper's algorithm does, where the Q-learning update happens across a union of an expert batch and an offline batch.

There was also a paper where behavioral cloning combined with RL has been used in autonomous driving [2], which does something similar to what this paper does, but in the online setting. This is very similar to TD3+BC though, and therefore it may not be a really big weakness to not cite this.

[1] Jonathan Chang, Masatoshi Uehara, Dhruv Sreenivas, Rahul Kidambi, Wen Sun; Mitigating Covariate Shift in Imitation Learning via Offline Data with Partial Coverage

[2] Yiren Lu, Justin Fu, ..., Shimon Whiteson, Dragomir Anguelov, Sergey Levine; Imitation is not Enough: Robustifying Imitation Learning with Reinforcement Learning for Challenging Driving Scenarios

**Questions:**

I didn't really have any big questions on this paper -- very solid!

**Details Of Ethics Concerns:**

None.

---

> ### Author Response · Authors · 2023-11-15
>
> Thanks for your positive comments and suggestions.
>
> **Weaknesses: missing citations**
>
> The discussion of these two works has already been added to the revised version.
> The discussion about BC-SAC [2] has been placed in section 3.3. The discussion about MILO [1] has been placed in Appendix B, where we provide a review of previous model-based methods and discuss our differences from them.
>
> [1] Mitigating Covariate Shift in Imitation Learning via Offline Data with Partial Coverage. NeurIPS’21
> [2] Imitation is not Enough: Robustifying Imitation Learning with Reinforcement Learning for Challenging Driving Scenarios. IROS’23

---

### Official Review · Reviewer_Yds8 · 2023-11-01

**Soundness:** 3 good
**Presentation:** 3 good
**Contribution:** 3 good
**Rating:** 6
**Confidence:** 4

**Summary:**

The paper addresses the challenge of Offline Imitation Learning (OIL), where an agent learns from both expert and sub-optimal demonstrations without further online interactions. Traditional studies in this domain rely heavily on high-quality behavioral data and falter when only low-quality, off-policy data is available. This research challenges that norm, asserting that even low-quality data can be beneficial for OIL. The authors propose a method that uses transition information from offline data to guide the policy towards states observed by experts, especially when reward signals are absent. They introduce an algorithm called Behavioral Cloning with Dynamic Programming (BCDP) that applies behavioral cloning to expert data and dynamic programming to unlabeled offline data. In tests, the BCDP algorithm outperforms many existing methods, showing a performance boost of over 40% even with purely random offline data.

**Strengths:**

The research presents a fresh perspective on offline imitation learning by challenging the conventional reliance on high-quality auxiliary data. Instead of seeing low-quality, off-policy data as a limitation, the authors innovatively harness its transition information to optimize objectives in states not observed by experts. The introduction of the BCDP algorithm, which combines behavioral cloning and dynamic programming, further underscores the paper's originality.

The quality of the research is evident in its comprehensive approach to the problem. Not only does the paper identify the challenge with low-quality auxiliary data, but it also offers a robust solution in the form of the BCDP algorithm. The empirical validation, where BCDP achieves state-of-the-art results on the D4RL benchmark across 14 tasks, attests to the method's efficacy and the overall quality of the research.

The paper lucidly articulates the challenges associated with offline imitation learning and the potential of low-quality data. The proposed BCDP algorithm is presented with clarity, making its methodology and implications easily understandable.

The research holds significant importance in the domain of imitation learning. By demonstrating that low-quality behavior data can be effectively leveraged, the paper breaks away from the traditional behavior quality assumption of auxiliary data, broadening the horizons of offline imitation learning. The potential extensions of BCDP, such as integrating it with model-based methods and addressing the existing imitation gap, highlight the paper's foundational contribution and its potential to pave the way for future advancements in the field.

**Weaknesses:**

The explanations for the experiment results, especially about the performance of BCDP, need more details.

One minor issue:
1. Table 1: the second-best result is not underlined.

**Questions:**

From Table 1, we can observe that BCDP performs very well for random or low-quality data. But when more expert knowledge is included, BCDP has inferior results than others. So, please
1. provide more details about the datasets, especially about the quality comparison regarding the experts.
2. explain why BCDP has less effective performance for non-random and human datasets.

---

> ### Author Response · Authors · 2023-11-15
>
> Thanks for your positive comments and valuable suggestions. Below are our responses to the concerns.
>
> **Q1: Details of data quality.**
>
> We introduced the data acquisition in Section 4.1, most of which comes directly from the D4RL benchmark. In addition to low-quality random exploration, the experiments also include multi-task data (umaze-v1, medium-v1, large-v1), early-stopped policy (-medium-v2), and human demonstrations (human-v1). We also provided quantitative comparisons in Appendix C.1.
>
>
> **Q2: Explanations on non-random and human datasets.**
>
> In this paper, we mainly focus on whether it is possible to remove previous assumptions on high-quality behavioral data and attempt to take advantage of the benefits of low-quality data. When the data contains some high-quality behavioral data, our method only uses the transition information in the data. This may be conservative, so its performance improvement is not as significant as in the setting with purely low-quality data. Fortunately, as suggested by reviewer uGnS, in this case, we can also combine previous research, such as DWBC or OTIL, to identify potential expert-similar data and further enhance performance. The corresponding experimental results are provided in Appendix C.6.
>
> In human data, the number of auxiliary data is limited (25 trajectories per task), which makes the transition information they can provide relatively limited. Therefore, performance improvement brought by BCDP is not very significant.
>
> **Weakness: One minor issue: Table 1: the second-best result is not underlined.**
>
> We have revised it in the new version.

---

> > ### Comment · Reviewer_Yds8 · 2023-11-22
> >
> > Thank the authors for their detailed explanation. After reading the reviews from other reviewers and the responses from the authors, I'd like to keep my current score unchanged.

---

### Meta-Review · Area_Chair_8HZz · 2023-12-05

**Metareview:**

The paper addresses the problem of offline imitation learning where the agent is given two sets of demonstrations: (i) a (small) set of expert data and (ii) a typically much larger dataset of suboptimal exploration data. The idea is to use the second dataset to learn how the world works and user the first dataset to learn about what the expert's goal is. The main algorithmic idea of the paper is to use an indicator reward on the expert dataset (defined below equation (8)) in combination with a batch RL algorithm for the "classic" setting (i.e. one that relies on rewards).

Strengths:
 - interesting setting

Weaknesses:
-  Proposition 1 (the main theoretical result of the paper) is not useful. I find the assumption that every state-action pair visited by the expert has high reward to be strange. If the expert is solving a goal, there will be some expert state-action pairs with high reward and some without.
 - I cannot seem to locate the proof of Proposition 1 in the paper.
 - The paper does not prove that divergence between the expert and imitator goes to zero as dataset size approaches infinity. This seems like a reasonable ask for an imitation learning algorithm (offline or not). I understand of course there are papers that don't do that, but I don't find that convincing.
- In any case, empirical results in a paper with essentially no theory should be extraordinarily good (they aren't).
- Empirical evaluation does not use rliable (https://github.com/google-research/rliable) or similar to summarise results. The reasons for doing so are well-known [1].
- Using the indicator reward as in Algorithm 1 seems problematic because the same state-action pair can be assigned reward zero or one depending on whether it happens to be sampled from the expert or exploration dataset. This is the same problem SQIL [2] has. In other words, algorithm 1 does not match equation (8).

[1] Agarwal et al. Deep Reinforcement Learning at the Edge of the Statistical Precipice

[2] Siddharth Reddy, Anca D. Dragan, and Sergey Levine. SQIL: imitation learning via reinforcement learning with sparse rewards

**Justification For Why Not Higher Score:**

This is a very weak paper. See weakness list in the meta-review.

**Justification For Why Not Lower Score:**

N/A

---

### Decision · Program_Chairs · 2024-01-16

Reject